# Combination of 20(R)-Rg3 and HUCMSCs Alleviates Type 2 Diabetes Mellitus in C57BL/6 Mice by Activating the PI3K/Akt Signaling Pathway

**DOI:** 10.3390/ijms262311469

**Published:** 2025-11-27

**Authors:** Zhengjie Zhou, Jingtong Zheng, Xiaoping Guo, Guoqiang Wang, Fang Wang, Xiaoting Meng

**Affiliations:** 1Department of Pathogen Biology, College of Basic Medical Sciences, Jilin University, Changchun 130021, China; 2Department of Histology & Embryology, College of Basic Medical Sciences, Jilin University, Changchun 130021, China

**Keywords:** 20(R)-Rg3, type 2 diabetes mellitus, HUCMSCs, insulin resistance

## Abstract

Type 2 diabetes mellitus (T2DM) is a global health challenge characterized by insulin resistance and pancreatic β-cell dysfunction. While human umbilical cord mesenchymal stem cells (HUCMSCs) show therapeutic potential, their efficacy can be limited by the harsh in vivo microenvironment. 20(R)-Rg3, a ginsenoside with anti-inflammatory and antioxidant properties, may enhance HUCMSCs’ function, but the combined effect and mechanism of this “cell-molecule” strategy remain unclear. This study aimed to investigate the therapeutic effects and underlying mechanisms of a combination therapy using 20(R)-Rg3 and HUCMSCs in a high-fat diet (HFD) and streptozotocin (STZ)-induced T2DM mouse model. Diabetic mice were treated with PBS, HUCMSCs alone, or HUCMSCs pre-treated with 20(R)-Rg3. Fasting blood glucose and body weight were monitored. Insulin resistance was assessed via oral glucose tolerance tests (OGTTs) and intraperitoneal insulin tolerance tests (IPITTs). Serum biochemical parameters (lipids, liver and kidney function, insulin, C-peptide) were analyzed. Histopathological examination (H&E, PAS) of the liver, kidney, and pancreas was performed, alongside immunofluorescence for islet hormones. Transcriptomic analysis (RNA-seq) was conducted on HUCMSCs with or without 20(R)-Rg3 pretreatment to elucidate potential signaling pathways. Results demonstrated that the combination significantly reduced hyperglycemia and improved insulin sensitivity more effectively than HUCMSCs alone. It also ameliorated dyslipidemia, enhanced liver and kidney function, promoted glycogen synthesis, and facilitated pancreatic islet “regeneration”. Transcriptomic analysis indicated that the synergistic effect is primarily mediated through activation of the PI3K/Akt signaling pathway. These findings suggest that 20(R)-Rg3 potentiates the therapeutic efficacy of HUCMSCs, providing a promising combinatorial strategy for T2DM treatment.

## 1. Introduction

Diabetes mellitus (DM) is recognized as a significant metabolic disorder affecting approximately 500 million individuals globally [1]. This prevalent condition has emerged as one of the primary challenges for global public healthcare systems, with its increasing prevalence posing serious implications for both individual health and societal healthcare resources [2]. Among the various forms of diabetes, type 2 diabetes mellitus (T2DM) is particularly prevalent, accounting for 90–95% of all DM cases [3]. T2DM arises from a complex interplay of insulin resistance and the impaired function of pancreatic beta cells, which disrupts insulin production and metabolism [4,5].

The liver, adipose tissue, and skeletal muscles are the three principal insulin-sensitive tissues that play critical roles in glucose homeostasis [6,7,8]. Dysfunction in these tissues can lead to dysregulated blood glucose levels, exacerbating complications associated with diabetes [9]. Chronic inflammation is a significant contributor to both insulin resistance and the progression of diabetic complications such as retinopathy, neuropathy, and vasculopathy [10,11]. Therefore, addressing inflammation is essential for comprehensive diabetes management [12].

Current treatment strategies for T2DM [13] primarily involve the use of oral antidiabetic medications, such as sulfonylureas, metformin, and thiazolidinediones, along with exogenous insulin injections. However, these approaches provide only temporary glycemic control and are associated with various adverse effects, including ametropia, subcutaneous nodules, diarrhea, and weight gain [14]. Moreover, prolonged administration of insulin may further impair endogenous β-cell function, complicating efforts to restore peripheral insulin sensitivity and alleviate symptoms associated with diabetic complications [15,16]. Consequently, there is an urgent need for novel therapeutic strategies.

Mesenchymal stem cell (MSC) therapy has emerged as a promising regenerative strategy for T2DM [17]. HUCMSCs have shown potential in improving glycemic control and β-cell function through their paracrine effects and immunomodulatory properties [18,19]. However, the therapeutic efficacy of MSCs is often limited by the harsh diabetic microenvironment, characterized by hyperglycemia and chronic inflammation, which compromises cell survival and function after transplantation. Therefore, strategies to enhance MSC resilience and function in vivo are crucial. 20(R)-Rg3, with its renowned anti-inflammatory and antioxidant properties, could potentially precondition HUCMSCs, creating a synergistic “cell-molecule” therapy that protects the cells and amplifies their therapeutic effects.

Ginseng, a traditional Chinese medicine, has demonstrated significant potential in the treatment of metabolic diseases over the past few decades [20]. The primary bioactive component of ginseng is ginsenoside Rg3, an active monomer extracted from this medicinal herb [21]. 20(R)-Rg3 exhibits a range of pharmacological properties, including anti-inflammatory [22], antidiabetic activity [23], and neuroprotective effects [24,25]. Specifically, 20(R)-Rg3 enhances insulin sensitivity by activating adenosine monophosphate-activated protein kinase (AMPK) and inhibits adipogenesis through peroxisome proliferator-activated receptor γ (PPARγ) [26,27]. Given AMPK’s central role in metabolic regulation, it represents a promising target for improving glucose metabolism and alleviating insulin resistance [28].

Emerging evidence further indicates that ginsenosides can simultaneously enhance renal function, bolster antioxidant capacity, exert anti-inflammatory effects, and modulate blood glucose levels [29], thereby linking metabolic and renoprotective actions within a single mechanistic framework [10].

In this study, we aimed to elucidate the mitigating effects of 20(R)-Rg3 in combination with HUCMSCs on T2DM and to explore its potential mechanisms of action. To achieve this, we first established a high-fat diet (HFD) and streptozotocin (STZ)-induced T2DM mouse model. A comprehensive evaluation was conducted to systematically assess the effects of 20(R)-Rg3 combined with HUCMSCs on T2DM, which included the measurement of T2DM-related serum biochemical indices, histopathological evaluations, and immunofluorescence staining. Subsequently, we analyzed the transcriptomic results following the pretreatment of HUCMSCs with 20(R)-Rg3 to predict the potential mechanisms underlying the remission of T2DM. This study posits that the combination of 20(R)-Rg3 and HUCMSCs may overcome the limitations of traditional pharmacological treatments and establish a new paradigm of synergistic intervention for T2DM through a “cell-molecule” approach.

## 2. Results

### 2.1. Identification of HUCMSCs and HFD/STZ-Induced T2DM Mouse Model

The cells exhibited adherent growth, displaying long spindles or multipolar morphologies, and formed swirling colonies within three days, indicating stable cell morphology (Figure 1A). Following 18 days of osteogenic induction, HUCMSCs transitioned from a spindle-shaped to a more cuboidal morphology, with numerous orange-red mineralized nodules visible upon Alizarin red-S staining (indicated by black arrows, Figure 1B). After 14 days of adipogenic induction, the cells adopted a round or oval shape and accumulated intracellular orange-red lipid droplets, as confirmed by positive oil red-O staining (indicated by red arrows, Figure 1C). Flow cytometric analysis of HUCMSCs surface markers revealed high expression levels of CD73 (99.27%), CD90 (99.96%), and CD105 (99.4%), and low expression of CD34 (0.85%), CD45 (0.84%), and HLA-DR (0.80%) (Figure 1D). These results indicate that HUCMSCs possessed typical morphological characteristics, exhibited complete osteogenic and adipogenic differentiation potential, and displayed a pure phenotypic profile, fulfilling the identification criteria established by the International Society for Cell and Gene Therapy (ISCT), thus making them suitable for subsequent experimental and clinical studies.

To evaluate the therapeutic effects of HUCMSCs on T2DM, we established a mouse model of T2DM using a combination of HFD and STZ administration. The results revealed that blood glucose levels in T2DM mice exceeded 20 mmol/L, whereas those in the Control group remained below 10 mmol/L, under both fasting and refed conditions (Figure 2A). The outcomes of OGTTs (Figure 2B), IPITTs (Figure 2C), serum C-P levels (Figure 2D), and pancreatic histology (Figure 2E) collectively indicated impaired insulin sensitivity, decreased islet volume, increased islet number (blue arrows), peri-islet vasodilation (red arrows), significant infiltration of inflammatory cells green arrows), and impaired serum insulin levels in the model mice. Furthermore, the mice exhibited classic symptoms of T2DM, including polyphagia, polydipsia, and polyuria. Taken together, these findings confirm the successful establishment of a T2DM animal model.

### 2.2. Histological Evidence That HUCMSCs Combined with 20(R)-Rg3 Reverses Insulin Resistance in T2DM

To evaluate the efficacy of the combined therapy, we first monitored body weight changes in both the Control and HFD groups. The results indicated that the body weight of mice in the HFD group reached a maximum of 29 ± 1.2 g, which was significantly higher than that in the Control group, with a statistically significant difference between the groups (** *p* < 0.01, Figure 3A). Subsequently, T2DM mice received tail vein injections of either HUCMSCs combined with 20(R)-Rg3, HUCMSCs alone, or PBS (as the control). Blood glucose levels were monitored every three days for three weeks (Figure 3B). Continuous glucose monitoring demonstrated that the combination of HUCMSCs and 20(R)-Rg3 significantly ameliorated hyperglycemia in T2DM mice, with blood glucose levels declining and stabilizing at 9.1 ± 2.80 mmol/L, which was significantly lower (* *p* < 0.05) than that in the HUCMSCs-alone group (13.73 ± 2.35 mmol/L). In contrast, blood glucose levels in T2DM control mice increased to 28.13 ± 1.60 mmol/L.

Peripheral insulin resistance is a key pathological feature of T2DM. This study investigated the effects of HUCMSCs combined with 20(R)-Rg3 on insulin sensitivity and blood glucose regulation in T2DM mice. After the final injection, OGTTs and IPITTs were performed. The OGTTs results indicated that the blood glucose levels in the combination treatment group were significantly lower than those in the T2DM group (10.1 ± 1.15 mmol/L, ** *p* < 0.01, Figure 3C), indicating substantially improved glucose tolerance. Furthermore, IPITTs results revealed that the blood glucose levels in the combination treatment group approached the normal range, at approximately 9.8 ± 2.06 mmol/L, which was significantly different from the T2DM group (* *p* < 0.05, Figure 3D).

HOMA-IR values are presented as mean ± SD. The blank group showed a basal index of 3.6 ± 1.6. In the T2DM-model group, HOMA-IR rose sharply to 7.3 ± 0.5 (** *p* < 0.01), indicating pronounced insulin resistance. HUCMSCs monotherapy reduced HOMA-IR to 5.8 ± 0.3 (* *p* < 0.05 vs. model), reflecting partial improvement in insulin sensitivity. Combination treatment (MSC + 20(R)-Rg3) further decreased HOMA-IR to 3.6 ± 0.49, significantly lower than MSC alone (** *p* < 0.01), demonstrating a synergistic and near-complete reversal of insulin resistance (Figure 3E).

HOMA-β values are expressed as mean ± SD. In the blank group, the index was 80.4% ± 2.3%, whereas it fell sharply in the T2DM-model group to 19.6% ± 0.7% (** *p* < 0.05), reflecting pronounced β-cell dysfunction. MSC monotherapy raised HOMA-β to 45.3% ± 1.5% (** *p* < 0.05 versus model), indicating partial recovery of insulin secretory capacity. Combination treatment (MSC + 20(R)-Rg3) further increased HOMA-β to 59.7% ± 3.2%, significantly higher than MSC alone (** *p* < 0.01), demonstrating a marked synergistic improvement in β-cell function (Figure 3F).

### 2.3. Serological Insights into the Attenuation of Insulin Resistance by HUCMSCs Combined with 20(R)-Rg3 in T2DM

Decreased TP levels may indicate impaired hepatic synthesis. TP levels were significantly lower in T2DM mice (40.7 ± 1.3 g/L) compared to the Control group (60.4 ± 3.6 g/L), suggesting severely compromised hepatic function. In contrast, treatment with HUCMSCs combined with 20(R)-Rg3 restored TP level to 57.2 ± 1.8 g/L, demonstrating a more pronounced therapeutic effect than HUCMSCs alone (Figure 4A). ALB, another crucial indicator of liver function, decreased to 20.4 ± 2.1 g/L but increased to 29.2 ± 4.1 g/L after combination treatment (Figure 4B), indicating significant improvement in hepatic synthetic function. Elevated AST and ALT levels, characteristic of impaired liver function in T2DM mice, were also reduced following combination treatment: AST decreased to 160 ± 9.6 U/L (Figure 4C) and ALT to 50 ± 6.4 U/L (Figure 4D), with significant differences between groups (** *p* < 0.01).

Dyslipidemia is a prevalent complication in T2DM. Treatment with HUCMSCs combined with 20(R)-Rg3 significantly reduced TC (2.5 ± 0.8 mmol/L, Figure 4E), TG (1.6 ± 0.4 mmol/L, Figure 4F), and LDL (0.5 ± 0.2 mmol/L, Figure 4H). Statistically significant differences were observed between the combination treatment and T2DM groups (** *p* < 0.01). Additionally, the treatment group exhibited a notable increase in HDL, approaching levels comparable to those of the Control group. Elevated serum creatinine levels typically indicate compromised renal function in T2DM mice. In the combination treatment group, BUN and serum Cr levels decreased to 8 ± 0.6 mmol/L, suggesting improved glomerular filtration rate and reduced tubular injury. A statistically significant difference was observed between the groups (** *p* < 0.01, Figure 4I,J).

### 2.4. In Vivo Imaging and Homing of HUCMSCs After Tail-Vein Injection

To investigate the biodistribution and homing behavior of HUCMSCs following systemic administration, cells were labeled with a near-infrared fluorescent dye (DiR) and injected into mice via the tail vein. Whole-body fluorescence imaging revealed a time-dependent redistribution of the labeled cells. Within the first hour, the majority of the signal was localized to the lungs, consistent with the first-pass entrapment of cells in the pulmonary microvasculature. However, by 24 h, a significant shift in fluorescence intensity was observed, with a marked accumulation of signal in the abdominal region, particularly within the liver (Figure 5).

### 2.5. HUCMSCs Combined with 20(R)-Rg3 Promote Glycogen Storage in T2DM Mice

Histopathological analysis of liver and kidney tissues from T2DM mice revealed significant structural alterations, as shown by H&E staining (Figure 6A). In control mice, liver tissues exhibited a regular arrangement of hepatocytes radiating from the central vein, with normal morphology and no evident pathology. Similarly, kidney tissues from Control mice displayed intact glomerular morphology, clear tubular structures, and normal cellular organization without apparent damage. In contrast, livers of T2DM mice showed lipid deposition, vacuolated hepatocytes, focal necrosis, disorganized cellular architecture (black arrows), and marked infiltration of mononuclear inflammatory cells. The kidneys of T2DM mice exhibited glomerular hypertrophy and hyperplasia, thickening of the glomerular basement membrane, increased protein deposition, and vacuolar degeneration of the renal tubules (red arrows). In the HUCMSCs treatment group, hepatic fatty degeneration and inflammatory infiltration were reduced (black arrows), and renal glomerular size was slightly decreased with attenuated vacuolar degeneration (red arrows). The combination treatment group showed near-normal hepatocyte arrangement and morphology (black arrows), significantly reduced inflammatory infiltration, and markedly ameliorated renal injury, with improved glomerular and tubular structures alongside reduced inflammation and fibrosis (red arrows).

PAS of liver and kidney tissues revealed significant pathological alterations in T2DM mice, reflecting the impact of diabetes on glycogen deposition and metabolic abnormalities (Figure 6B). Livers of Control mice showed moderate, light purple-red glycogen distribution with clearly visible nuclei. Kidneys exhibited minimal glycogen deposition in glomeruli and tubuli, with light staining and clear cellular structures. T2DM livers displayed excessive glycogen accumulation, hepatocyte swelling, and vacuolar degeneration. T2DM kidneys showed heightened glycogen deposition in glomeruli and tubuli, accompanied by tubular atrophy, basement membrane thickening, and fibrosis. HUCMSCs treatment reduced hepatic glycogen content and alleviated vacuolation, as well as renal tubular atrophy and fibrosis. The combination treatment group showed a significant reduction in hepatic glycogen, approaching normal levels (Figure 6C), and markedly decreased glycogen deposition in renal tissues (Figure 6D).

HUCMSCs combined with 20(R)-Rg3 synchronously reverse gluconeogenesis, activate glycolysis, and suppress inflammation at the transcriptional level. Subsequently, we evaluated the effect of combination therapy on key enzymes and inflammatory factors related to glucose metabolism. Glucokinase (GCK), arginase-1 (Arg-1), phosphoenolpyruvate carboxykinase (PEPCK), glucose-6-phosphatase (G6Pase), and peroxisome proliferator-activated receptor gamma coactivator 1-alpha (PGC-1α) are molecules closely associated with hepatic gluconeogenesis and play crucial roles in regulating glucose metabolism, insulin resistance, and metabolic diseases such as diabetes mellitus. RT-qPCR analysis indicated that mRNA expression of G6Pase, a critical gluconeogenic enzyme, was significantly decreased in T2DM mice but increased after both HUCMSCs and combination treatments (** *p* < 0.01; Figure 7A). GCK, a key enzyme in the glycolytic pathway primarily expressed in the liver, is integral to glucose uptake and metabolism. GCK expression, typically reduced in T2DM, was elevated in the combination group (Figure 7B). Arg-1 mRNA, associated with anti-inflammatory and metabolic functions, was significantly enhanced by the combination treatment (Figure 7C). PEPCK and PGC-1αexpression, key regulators of gluconeogenesis and metabolic homeostasis, were more effectively restored in the combination treatment group (Figure 7D,E). Furthermore, combination treatment significantly reduced mRNA levels of the inflammatory cytokines TNF-α and IL-1β (Figure 7F,G). Collectively, these findings indicate that HUCMSCs combined with 20(R)-Rg3 improve insulin signaling, promote glycogen synthesis, restore glucose homeostasis in the liver, and enhance glycolysis in T2DM mice.

### 2.6. HUCMSCs Combined with 20(R)-Rg3 Promote Insulin Secretion and Islet “Regeneration”

Pancreatic islet morphology and function were assessed using double-label immunofluorescence staining. In T2DM, insulin release from pancreatic β-cells is impaired. Consistent with this, HFD/STZ-induced islet damage in T2DM mice led to compromised insulin production. Immunofluorescence analysis revealed deteriorated islet structure in diabetic mice. As shown in Figure 8A, dual staining for glucagon (green) and insulin (red) showed a reduction in both glucagon-positive α-cells and insulin-positive β-cells in T2DM mice.

Quantitative analysis demonstrated that the number of islets per pancreatic section was significantly reduced in the T2DM group compared to normal controls. Treatment with HUCMSCs combined with 20(R)-Rg3 resulted in a significant increase in islet number compared to the T2DM group (Figure 8B). The ratio of insulin-positive cells to total pancreatic tissue area was approximately 4% in Control mice, which declined to about 1% in T2DM mice. The HUCMSCs-alone group showed a modest increase in this ratio, while the combination treatment group exhibited a significant increase, with a statistically significant difference (** *p* < 0.01) compared to the T2DM group (Figure 8C). The normalization of blood glucose levels was accompanied by dynamic changes in insulin and glucagon expression, suggesting possible “regeneration” and functional transformation of pancreatic islet cells following combination therapy.

### 2.7. Rg3 Exerts a Protective Effect Against Spontaneous Apoptosis in HUCMSCs

Flow-cytometric analysis with Annexin-V-FITC/PI staining revealed the following apoptotic indices. Results showed that the control group had an Apoptosis rate of (3.9 ± 0.5)%, with cells in good condition and a low level of apoptosis (Figure 9A). In the T2DM model group, the apoptosis rate was (36.1 ± 0.3)%, significantly higher than that of the control group, indicating successful model establishment and significant cellular damage (Figure 9B). The HUCMSCs monotherapy group exhibited an apoptosis rate of (15 ± 0.1)%, showing a decreasing trend compared to the model group and suggesting MSCs possess certain anti-apoptotic effects (Figure 9C). The HUCMSCs combined with 20(R)-Rg3 therapy group showed an apoptosis rate of (11.7 ± 0.5)%, which was lower than both the model group and the monotherapy group (Figure 9D). This indicates that HUCMSCs combined with Rg3 exhibit a synergistic effect in inhibiting apoptosis, with efficacy superior to that of monotherapy. In summary, HUCMSCs combined with 20(R)-Rg3 therapy significantly reduced the apoptosis rate, demonstrating enhanced cytoprotective effects.

### 2.8. Transcriptome Differential Expression Analysis

In this study, we utilized HUCMSCs as a model system to identify key signaling pathways and candidate genes associated with diabetes, with the aim of discovering potential therapeutic targets. HUCMSCs were treated with 20(R)-Rg3 for a duration of six hours. Both the 20(R)-Rg3-treated group and the untreated control group included three biological replicates. Total RNA was extracted from each group and subjected to transcriptome sequencing.

Principal component analysis (PCA) of the gene expression matrix showed high intra-group consistency and clear separation between the treated and control groups, indicating distinct transcriptomic profiles (Figure 10A). Differential expression analysis identified 69 genes with significant differential expression (|log_2_FC| > 1, FDR < 0.05; Figure 10B). Among these, TXNIP was the most significantly downregulated gene (FDR = 1.30 × 10^−129^), while DUSP1 was the most significantly upregulated gene after 20(R)-Rg3 treatment (FDR = 1.30 × 10^−129^) (Figure 10C,D). These findings provide valuable insights into the molecular mechanisms through which 20(R)-Rg3 may influence diabetes-related pathways in HUCMSCs.

### 2.9. Functional Enrichment Analysis of Differentially Expressed Genes

GO enrichment analysis of the differentially expressed genes (DEGs) revealed significant enrichment in biological processes, molecular functions, and cellular components. The top-enriched GO terms included protein binding, regulation of type B pancreatic cell proliferation, and bicellular tight junctions. These findings suggest that the DEGs are functionally linked to essential processes related to pancreatic function and intercellular communication (Figure 11A).

KEGG pathway enrichment analysis further revealed that the DEGs were significantly enriched in 94 pathways (FDR < 0.01), including cytokine–cytokine receptor interactions, Jak–STAT signaling pathway, and tight junction pathways (Figure 11B). These findings suggest that 20(R)-Rg3 treatment may influence multiple signaling cascades associated with immune responses, inflammation, and the integrity of cell–cell junctions.

Further screening of the KEGG enrichment results identified ten significantly enriched pathways previously reported to be involved in the pathogenesis and progression of diabetes, either through direct regulation of glucose metabolism or indirect modulation of inflammatory and immune responses (Figure 11C). This supports the hypothesis that 20(R)-Rg3 acts through diabetes-associated signaling pathways. To better understand the functional roles of these DEGs, a regulatory interaction network was constructed, revealing that many DEGs occupy central positions and may act as core regulators in diabetes-related pathways (Figure 11D). Additionally, protein–protein interaction (PPI) analysis further indicated that numerous candidate genes interact directly or indirectly, forming a dense interaction network that suggests functional cooperation in disease processes (Figure 11E).

Expression analysis of diabetes-associated DEGs such as CXCL8, IL7R, and NR4A1 showed significant alterations after 20(R)-Rg3 treatment. These genes are known to be associated with insulin signaling, glucose homeostasis, and inflammatory regulation (Figure 11F). Collectively, these findings suggest that 20(R)-Rg3 induces a coordinated transcriptional response in HUCMSCs, potentially influencing key mechanisms in diabetes development and progression.

## 3. Discussion

The core pathological mechanisms of T2DM primarily involve defective pancreatic β-cell function and insulin resistance, collectively leading to a chronic hyperglycemic state. This metabolic dysregulation not only disrupts glucose homeostasis but also contributes to multi-system complications, including cardiovascular, renal, and neurologic disorders [30]. Metabolomics analyses have recently revealed that microbial-derived butyrate predicts postprandial glucose control independently of body weight [31,32]. Furthermore, the diabetic foot ulcer (DFU) is not merely a passive sequel to chronic hyperglycemia but functions as an active inflammatory focus. Persistent wound-derived cytokines spill into the circulation, amplify systemic inflammation, and establish a self-sustaining cycle that hastens metabolic decline [33]. According to the International Diabetes Federation (IDF), the global prevalence of T2DM has exceeded 500 million, with complications such as nephropathy and cardiovascular disease posing significant mortality risks [31]. Current treatment strategies primarily focus on glycemic control and include metformin (improving IR), sulfonylureas (stimulating insulin secretion), and GLP-1 receptor agonists (suppressing appetite and protecting β-cells) [32]. However, these therapies have notable limitations: they cannot reverse the irreversible damage to pancreatic β-cells (e.g., apoptosis induced by chronic hyperglycemia); their efficacy is limited in patients with severe IR; and they are associated with adverse effects such as hypoglycemia and gastrointestinal disturbances. Moreover, these interventions are not curative, typically require lifelong administration, and 30–40% of patients eventually experience therapeutic failure [34,35]. Consequently, there is an urgent need for innovative therapeutic strategies capable of restoring pancreatic islet function, ameliorating IR, and mitigating inflammation. The combination of 20(R)-Rg3 with HUCMSCs represents a promising approach to address this unmet need.

HUCMSCs, derived from Wharton’s jelly of the umbilical cord, constitute a therapeutic strategy centered on tissue ‘repair and regulation’. They have gained attention as a cell-based therapy for T2DM due to their low immunogenicity, accessibility, multidirectional differentiation potential, and paracrine functions. The mechanisms by which HUCMSCs exert their effects on T2DM can be summarized in three key points: (1) HUCMSCs promote the “regeneration” of damaged pancreatic cells through “induced differentiation” wherein they differentiate into islet-like β-cells and secrete insulin within a conducive microenvironment, or via paracrine support, in which they release cytokines such as vascular endothelial growth factor (VEGF) and hepatocyte growth factor (HGF) that enhance the proliferation and survival of damaged islet β-cells, ultimately improving both the quantity and functionality of β-cells [36]. (2) HUCMSCs mitigate insulin resistance (IR)by regulating adipocyte metabolism, which includes reducing the release of inflammatory factors such as TNF-α and IL-6, and by enhancing hepatic glucose and lipid metabolism, promoting glycogen synthesis while inhibiting gluconeogenesis [37]. (3) HUCMSCs inhibit the over-activation of immune cells and decrease the release of inflammatory factors through the secretion of TGF-β and prostaglandin E2 (PGE2). Concurrently, melatonin alleviates oxidative stress-induced damage to pancreatic islets by upregulating antioxidant enzymes such as superoxide dismutase (SOD). Similarly, studies have shown that melatonin attenuates oxidative stress-induced mitochondrial dysfunction in cardiomyocytes. Together, these findings suggest that circulating melatonin may protect mitochondrial integrity under ischemic conditions, as evidenced by its inverse correlation with cardiac injury markers [38]. Animal studies have demonstrated that HUCMSCs transplantation can lower fasting blood glucose levels, elevate insulin levels, and reduce islet fibrosis-indicative of β-cell repair in T2DM model mice. Additionally, murine data demonstrate that, albeit with small sample sizes, have confirmed that glycated hemoglobin (HbA1c) levels and insulin dosages in T2DM patients can decrease significantly within a short time frame (12–24 weeks) following HUCMSCs transplantation [17]. Although the PCR data demonstrate consistent transcriptional changes in the target genes, these observations were not corroborated at the protein or histological level in the tissue sections. Future work will employ immunohistochemistry and/or in situ hybridization to validate the PCR results spatially and qualitatively. We acknowledge that these findings are derived from preclinical models, and further validation in human cohorts is necessary to confirm their translational relevance.

Rg3 is a promising natural drug candidate focused on regulation and protection, a rare active ingredient primarily sourced from ginseng. This naturally occurring small molecule compound exhibits multiple pharmacological effects, including anti-inflammatory, antioxidant, and metabolic enhancement properties. Its mechanism of action in T2DM centers on improving the microenvironment and enhancing metabolic regulation [39]. Rg3 activates the AMPK pathway, a key molecule in the regulation of cellular energy metabolism, promoting adipocyte catabolism while inhibiting hepatic gluconeogenesis. Concurrently, it upregulates the expression of insulin receptor substrate (IRS-1), thereby enhancing insulin signaling and improving the efficiency of insulin-receptor binding. Additionally, Rg3 inhibits chronic inflammation and protects the islet microenvironment. Furthermore, Rg3 scavenges ROS, upregulates the expression of the antioxidant factor Nrf2, and reduces β-cell apoptosis induced by oxidative stress. Available evidence from animal experiments indicates that Rg3 intervention can effectively reduce blood glucose levels, improve glucose tolerance, and decrease inflammatory infiltration of pancreatic tissue in T2DM mice [40].

### Synergistic Mechanisms of HUCMSCs and 20(R)-Rg3: Toward a “1 + 1 > 2” Effect

The core pathology of T2DM encompasses a triad of β-cell failure, IR, and chronic inflammation. HUCMSCs demonstrate the capacity to repair β-cells and modulate immune responses; however, their efficacy is often compromised in the high-glucose and high-inflammation microenvironment that follows transplantation, resulting in low survival rates and functional inhibition [41,42]. In contrast, 20(R)-Rg3 improves the tissue microenvironment through anti-inflammatory and antioxidant actions but does not directly regenerate β-cells or increase their numbers. In this study, preconditioning HUCMSCs with 20(R)-Rg3 created a complementary strategy that leverages the strengths of both agents. An RG3-monotherapy group was not included; therefore, the synergy index (quantifying the combination effect relative to the sum of individual effects) could not be formally calculated. The observed superiority of the MSCs + Rg3 group is reported as an additive or greater-than-additive effect. A study limitation is the absence of an Rg3-alone cohort, which precludes definitive confirmation that the combination yields true synergy (1 + 1 > 2). Full 2 × 2 factorial studies that include Rg3-monotherapy, HUCMSC-monotherapy, combination, and vehicle groups will be required to quantify the exact interaction index and to distinguish synergy from simple additivity.

The two therapies exhibit synergistic effects through overlapping molecular targets. HUCMSCs promote active tissue repair and regulate IR-related pathways, while 20(R)-Rg3 optimizes the microenvironment, reduces inflammation, and improves insulin sensitivity, thereby enhancing HUCMSCs’ survival and function. HUCMSCs may ameliorate IR via paracrine-mediated AMPK activation. Both HUCMSCs and 20(R)-Rg3 synergistically regulate glucose and lipid metabolism: HUCMSCs activate AMPK indirectly through secreted factors, while 20(R)-Rg3 directly activates AMPK. Furthermore, HUCMSCs inhibit nuclear factor kappa B (NF-κB) through transforming growth factor-β (TGF-β), and 20(R)-Rg3 directly suppresses the phosphorylation of NF-κB p65, resulting in more effective anti-inflammatory effects. RNA-seq analysis of Rg3-primed HUCMSCs revealed significant enrichment of PI3K/Akt pathway genes, suggesting this axis as a putative mechanism that warrants direct validation in target tissues. While transcriptomic data point to PI3K/Akt enrichment, causal evidence in liver, muscle, adipose, and pancreatic tissue—such as phosphorylation status of Akt/PI3K and inhibitor-rescue studies—remains to be established in future work.

## 4. Materials and Methods

### 4.1. Animals and Treatment

Eight-week-old male C57BL/6J mice (16–20 g) were procured from Changchun Yisi Laboratory Animal Technology Co., Ltd. (SPF-grade, Certificate No. SCXK(JI)-2020-0002) and housed in the Laboratory Animal Center of the School of Basic Medical Sciences at Jilin University. The mice were maintained under controlled conditions (temperature: 20 ± 5 °C; humidity: 50 ± 10%; 12-h light/dark cycle) with ad libitum access to food and water. The study was conducted according to the guidelines of the Declaration of Helsinki and approved by the Institutional Review Board (or Ethics Committee) of Jilin University (protocol code No.2025-616 and approval date: 26 July 2025).

Animals were randomized using simple computer-generated numbers, and the treatment allocation was carried out by an investigator who was not involved in outcome assessment and remained blinded to group assignment. All mice were categorized into two groups: the T2DM group, which was given an HFD for ten weeks, and the “standard-diet control (SDC)” group, denoted as the “Control” group, which was fed a standard chow diet. After 10 weeks of high-fat diet feeding, mice were fasted for 6 h on day 70 of the experiment. A single intraperitoneal injection of streptozotocin (120 mg kg^−1^) was administered to the high-fat diet group to induce T2DM. One week post-STZ injection, an OGTT was performed after 12-h fasting (with free access to water). Glucose was administered orally (2 mg/kg body weight), and diabetes was confirmed based on the following criteria: fasting (0 min) blood glucose ≥ 7.8 mmol/L and 120 min blood glucose ≥ 11.1 mmol/L [43]. Fasting blood glucose levels were monitored daily through tail vein sampling using a glucometer. Mice exhibiting fasting blood glucose levels of ≥ 16.7 mmol/L were deemed to have successfully established a T2DM model.

High-fat diet (HFD) details: Exact caloric density: 4.7 kcal g^−1^ (19.7 kJ g^−1^); Macronutrient composition: 45% kcal from fat (lard/soybean oil 4:1), 20% kcal from protein (casein), 35% kcal from carbohydrates (sucrose/corn starch 1:2); Micronutrients: AIN-93G mineral and vitamin mix at 100% recommended level, plus 0.25% choline bitartrate and 0.014% tert-butylhydroquinone antioxidant; Physical form: pelleted, irradiated, stored at 4 °C and used within 4 weeks of manufacture.

In experimental study, mice were divided into four groups, each consisting of 10 mice: (1) The Control group, which received 200 μL of PBS injected via the tail vein; (2) The T2DM group, which also received 200 μL of PBS through the tail vein; (3) The HUCMSCs group, in which T2DM mice were injected weekly with 1 × 10^6^ HUCMSCs suspended in 200 μL of PBS via tail vein injection; and (4) The HUCMSCs + Rg3 group, where T2DM mice received weekly injections of 1 × 10^6^ HUCMSCs that had been pre-treated with 40 μM Rg3 for three days prior to administration (injected for three weeks) 20(R)-Rg3 (≥98%) dissolved in DMSO to prepare a 10 mM stock solution. When diluted to a 40 μM concentration, the final DMSO concentration is ≤0.1%. Additionally, the T2DM mouse model was established, and the schedule for pharmacological intervention is illustrated in Figure 12.

### 4.2. Oral Glucose Tolerance Tests (OGTTs)

After a 12-h overnight fast with access to water, the mice were weighed and administered a glucose solution (Kelun^®^, Chengdu, China) (2.0 g/kg) via oral gavage. Blood samples were collected from the tail vein at 0, 15-, 30-, 60-, 90-, and 120-min post-administration, and blood glucose levels were measured using a glucometer (Accu-Chek Active^®^, Shanghai, China, Product ID: 10030009783463).

### 4.3. Intraperitoneal Insulin Tolerance Tests (IPITTs)

Following a 12-h fasting period, mice were administered an intraperitoneal injection of insulin (Brand Name: NovoRapid^®^, Novo Nordisk, Denmark, PME650P, Rapid-acting insulin) at a dosage of 0.5 U/kg. Blood glucose levels were monitored at 0, 15-, 30-, 60-, 90-, and 120-min post-injection through tail vein sampling using a glucometer.

### 4.4. Culture for the Identification of HUCMSCs

HUCMSCs were obtained from Sciencell Research (ScienCell Research Laboratories, Carlsbad, CA, USA, Cat#7530) (Cells are for research purposes only). For phenotypic characterization, cultured HUCMSCs were analyzed using flow cytometry. Cells were washed and resuspended in PBS staining buffer at a density of 1 × 10^6^ cells/mL. The following fluorescently labeled antibodies were utilized: MSC markers (CD73-PE, CD90-FITC, and CD105-PE), hematopoietic markers (CD34-PE and CD45-FITC), and the MHC class II marker (HLA-DR-FITC) (all antibodies sourced from BD Biosciences, Shanghai, China). Gate-setting strategy: FSC-A vs. SSC-A: Exclude de-bris and dead cells. FSC-H vs. FSC-A: Exclude double-celled organisms. 7-AAD: Exclude dead cells (7-AAD^−^). Single-stain tubes: Used for compensation adjustment (single fluorescent antibody staining per tube). Cells were incubated with antibodies for 30 min at 2–8 °C in the dark, washed twice with PBS, and subsequently resuspended in 200 μL of PBS for analysis using FlowJo software (V9.01).

### 4.5. Osteogenic Differentiation Assay

HUCMSCs were seeded in 24-well plates at a density of 4 × 10^4^ cells per well. Once the cells reached 70–80% confluence, the medium was replaced with osteoblast induction medium (SUPERCULTURE, Cat#6114541). After 14 to 18 days of differentiation, depending on the actual differentiation status, the cells were fixed and stained with alizarin red to detect calcium deposition.

### 4.6. Adipogenic Differentiation Assay

HUCMSCs were seeded in 12-well plates at a density of 1 × 10^5^ cells per well. Upon reaching 70–80% confluence, the cells were maintained in mesenchymal stem cell culture medium. When confluence reached 80–90%, the culture medium was replaced with adipogenic differentiation medium (SUPERCULTURE, Cat#6114531). Following a differentiation period of 10 to 21 days, depending on lipid accumulation, the cells were fixed and stained with Oil Red O to visualize lipid droplets.

### 4.7. Biochemical Sampling and Analysis

At the end of the experimental period, mice were subjected to a 12-h fasting period prior to blood collection. Blood was collected from the retro-orbital sinus of anesthetized mice using heparinized capillary tubes. Following this, blood samples were centrifuged at 4 °C at 3500 rpm for 10 min to isolate the serum. The serum was subsequently analyzed for the following parameters: (1) liver function indicators, including total protein (TP), albumin (ALB), aspartate aminotransferase (AST), and alanine aminotransferase (ALT); (2) lipid profile metrics, such as total cholesterol (TC), triglycerides (TG), high-density lipoproteins (HDL), and low-density lipoproteins (LDL); (3) kidney function markers, specifically creatinine (Cr) and blood urea nitrogen (BUN) levels; and (4) metabolic markers, which included serum C-peptide (C-P) and insulin concentration (INS), both quantified using ELISA.

### 4.8. Real-Time Quantitative Polymerase Chain Reaction (RT-qPCR)

Using TRIzol reagent, reverse transcription was conducted to generate complementary DNA (cDNA), which was subsequently analyzed through quantitative RT-qPCR, with GAPDH serving as the internal control (RNA input: 200 ng per 20 µL reverse-transcription reaction (final concentration 10 ng µL^−1^). Quantification method: relative expression was calculated with the 2^−ΔΔCT^ method. Cycling conditions for every assay: 95 °C for 3 min; 40 cycles of 95 °C for 10 s, 60 °C for 30 s; melt-curve 95 °C for 15 s, 60 °C for 1 min, ramp to 95 °C at 0.3 °C s^−1^. All primers listed in Table 1 were synthesized by General Biologicals in Anhui, China.

### 4.9. Histological Analysis

Tissues from the liver, pancreas, and kidneys were preserved in a 4% paraformaldehyde solution for an overnight period. Following this, the samples underwent a series of graded alcohol treatments before being embedded in paraffin. Thin sections, each 5 μm thick, were prepared and stained with hematoxylin-eosin (H&E) to evaluate overall morphology, while periodate-Schiff staining (PAS) was employed to identify glycogen.

H&E staining: (1) Deparaffinise 5 µm sections in xylene (2 × 5 min) and rehydrate through graded ethanol (100%, 95%, 70%, 30%, 2 min each); (2) Rinse in tap water (2 min), stain with Mayer’s hematoxylin (Sigma-Aldrich, MHS16, St. Louis, MO, USA) for 5 min at room temperature (RT); (3) Wash in running tap water (5 min), differentiate in 1% acid ethanol (1 s), rinse in tap water (1 min); (4) Blue in 0.2% ammonia water (30 s), wash in tap water (2 min); (5) Counterstain with 1% eosin Y (Sigma-Aldrich, 318906) for 1 min, dehydrate through graded ethanol, clear in xylene and mount with DPX.

PAS: (1) Deparaffinise and rehydrate as above; (2) Oxidize in 0.5% periodic acid (Sigma-Aldrich, 395413) for 10 min at RT, rinse in distilled water (3 × 1 min); (3) Immerse in Schiff’s reagent (Sigma-Aldrich, 3952016) for 15 min in the dark, wash in lukewarm tap water (10 min); (4) Counterstain nuclei with Mayer’s hematoxylin (2 min), blue as above, dehydrate, clear and mount.

### 4.10. Immunofluorescence Analysis

Pancreatic sections were subjected to immunostaining to detect the expression of insulin and glucagon in β-cells, using specific primary antibodies followed by fluorescent secondary antibodies. The nuclei were counterstained with DAPI.

Immunofluorescence staining: (1) Deparaffinise 5 µm pancreatic sections in xylene (2 × 10 min) and rehydrate through graded ethanol (100%, 95%, 70%, 50%, 2 min each); (2) Antigen retrieval: heat sections in 10 mM sodium citrate (pH 6.0) at 95 °C for 20 min, cool to RT, rinse in PBS (3 × 5 min); (3) Block with 5% normal goat serum in PBS-T (0.1% Tween-20) for 1 h at RT; (4) Incubate overnight at 4 °C with primary antibodies diluted in blocking buffer: guinea-pig anti-insulin (1:200; Dako A0564, Shanghai, China); rabbit anti-glucagon (1:400; Abcam ab92517, Shanghai, China); (5) Wash in PBS-T (3 × 5 min), incubate with Alexa-Fluor-conjugated secondary antibodies (1:500; Invitrogen A-11073 and A-11034, Shanghai, China) for 1 h at RT in the dark; (6) Counterstain nuclei with DAPI (1 µg mL^−1^, 5 min), wash in PBS (3 × 5 min); (7) Mount with ProLong Gold antifade medium (Thermo Fisher P36930, Waltham, MA, USA) and image with a 20× objective on a Zeiss LSM 880 confocal microscope (Shanghai, China).

### 4.11. Flow Cytometry Protocol for Apoptosis Detection (Annexin V-FITC/PI Staining)

Apoptosis was assessed by flow cytometry using an Annexin V-FITC/PI double-staining assay. The MIN6 mouse pancreatic β-cell line was first revived and cultured for 24 h in a freshly prepared stimulation medium containing 33 mmol/L glucose and 200 μg/mL AGEs to allow for adherence. The medium was then switched to serum-free medium, followed by a 24-h stimulation period; it is noted that for MIN6 cells, the stimulation medium contained only 33 mmol/L glucose and AGEs to avoid hyperosmolarity-induced cell death. The experimental design included the following groups: a Blank Control cultured in normal medium; a Model group treated with high glucose, AGEs, and inflammatory factors for 24 h; an MSC group which, after modeling, was switched to normal medium and co-cultured with MSCs for 24 h; and an MSC + Rg3 group which, after modeling, was switched to a medium containing 40 µM 20(R)-Rg3 and co-cultured with MSCs for 24 h. Following treatments, cells were harvested at a density of 1 × 10^6^ cells per sample, washed twice with cold PBS, and centrifuged at 300× *g* for 5 min at 4 °C with the supernatant discarded after each wash. The cell pellet was then resuspended in 100 μL of 1× Annexin V binding buffer, followed by the addition of 5 μL of Annexin V-FITC and 5 μL of propidium iodide (PI, 50 μg/mL). The staining mixture was incubated for 15 min at room temperature in the dark. Immediately following incubation, 400 μL of 1× binding buffer was added to each tube, and the samples were analyzed using a flow cytometer. Fluorescence was detected with FITC at 530/30 nm (FL1), and PI at 610/20 nm (FL2), and a minimum of 10,000 events were gated per sample. Data interpretation defined the cell populations as follows: Annexin V^−^/PI^−^ as viable cells, Annexin V^+^/PI^−^ as early apoptotic cells, Annexin V^+^/PI^+^ as late apoptotic or necrotic cells, and Annexin V^−^/PI^+^ as necrotic cells (typically a rare population).

### 4.12. RNA Sequencing (RNA-Seq)

#### 4.12.1. Sample Quality Control

RNA sequencing was conducted on three cell groups: the Control group, the HUCMSCs group, and the HUCMSCs + Rg3 group, where HUCMSCs were preconditioned with 40 μM Rg3 for 3 days before harvest for RNA-seq or for transplantation. No washout was performed; cells were collected directly after the 3-day exposure. Differentially expressed genes (*P* < 0.05) were analyzed using Gene Ontology (GO) enrichment analysis and Kyoto Encyclopedia of Genes and Genomes (KEGG) pathway annotation. The RNA Nano 6000 Assay Kit from the Bioanalyzer 2100 system (Agilent Technologies, Santa Clara, CA, USA) was utilized to evaluate the integrity of RNA.

#### 4.12.2. Library Preparation for Transcriptome Sequencing

Total RNA was utilized as the input material for RNA sample preparations (Concentration: ≥100 ng/µL). Briefly, mRNA was purified from total RNA using poly-T oligo-attached magnetic beads. Fragmentation was conducted using divalent cations at elevated temperatures in the First Strand Synthesis Reaction Buffer (5X). The first strand of cDNA was synthesized using a random hexamer primer and M-MuLV Reverse Transcriptase (RNase H-). Subsequently, second-strand cDNA synthesis was performed using DNA Polymerase I and RNase H. Remaining overhangs were converted into blunt ends through exonuclease/polymerase activities. Following the adenylation of the 3′ ends of DNA fragments, adaptors with a hairpin loop structure were ligated to prepare for hybridization. To select cDNA fragments preferentially ranging from 370 to 420 bp in length, the library fragments were purified using the AMPure XP system. PCR was then performed with Phusion High-Fidelity DNA polymerase, universal PCR primers, and an Index (X) Primer. Finally, the PCR products were purified using the AMPure XP system, and library quality was assessed on the Agilent Bioanalyzer 2100 system.

#### 4.12.3. Clustering and Sequencing (Novogene Experimental Department)

The clustering of the index-coded samples was performed using a cBot Cluster Generation System with the TruSeq PE Cluster Kit v3-cBot-HS (Illumina, San Diego, CA, USA), following the manufacturer’s instructions. After the cluster generation, the library preparations were sequenced on an Illumina NovaSeq platform, resulting in the generation of 150 bp paired-end reads. Subsequently, differential gene analysis was conducted on the results obtained from the aforementioned experiment.

### 4.13. Statistical Analysis

All data are presented as mean ± standard error of the mean (mean ± SEM). Statistical analyses were performed using GraphPad Prism 8.0 software. Differences between the two groups were evaluated using Student’s *t*-test, while multiple group comparisons were analyzed through one-way ANOVA followed by post hoc tests. Statistical significance was defined as * *p* < 0.05, ** *p* < 0.01, *** *p* < 0.001, and **** *p* < 0.0001.

## 5. Conclusions

Combination therapy with ginsenoside 20(R)-Rg3 and HUCMSCs significantly reduced blood glucose levels, ameliorated islet damage, boosted the population of islet β-cells, improved hepatic and renal glucose metabolism, and enhanced insulin sensitivity in a mouse model of T2DM. These beneficial effects are likely mediated through activation of the PI3K/Akt signaling pathway.

While this study demonstrates promising results, several limitations should be considered. First, the research was conducted in an animal model, which may not fully recapitulate human T2DM pathophysiology. Second, the precise molecular mechanisms underlying the synergistic effects between HUCMSCs and 20(R)-Rg3 require further elucidation, particularly in terms of in vivo cell tracking and long-term safety. Third, the dose–response relationship and optimal treatment duration for the combination therapy remain to be determined. Finally, clinical studies are necessary to validate the efficacy and safety of this novel combination approach in human patients.

## Figures and Tables

**Figure 1 ijms-26-11469-f001:**
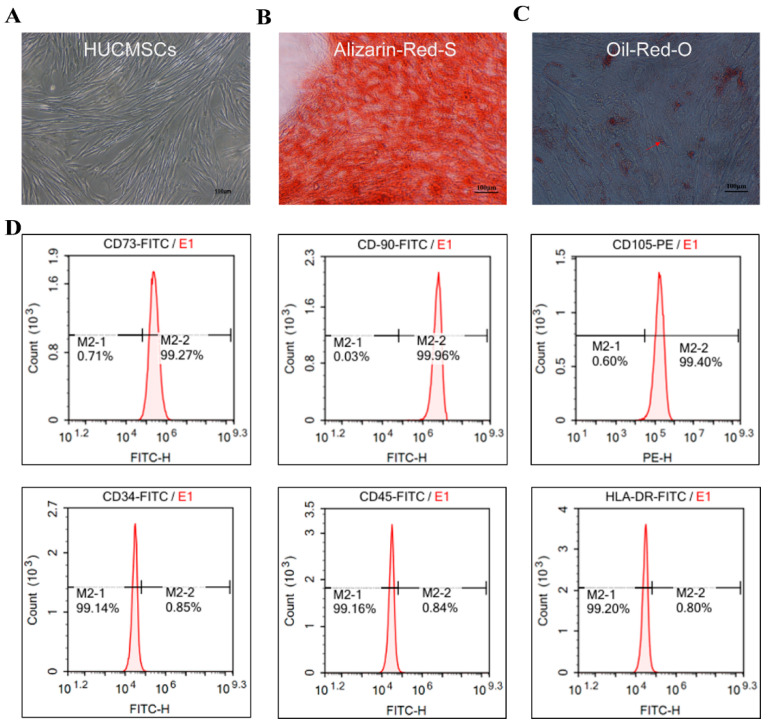
Identification of HUCMSCs. (**A**). HUCMSCs. (**B**). HUCMSCs and Osteogenic Differentiation. (**C**). HUCMSCs and Lipogenic Differentiation: red arrows (intracellular orangered lipid·droplets). Images were captured at a magnification of 100× (scale bar = 100 μm). (**D**). Flow-cytometric analysis of HUCMSCs surface markers.

**Figure 2 ijms-26-11469-f002:**
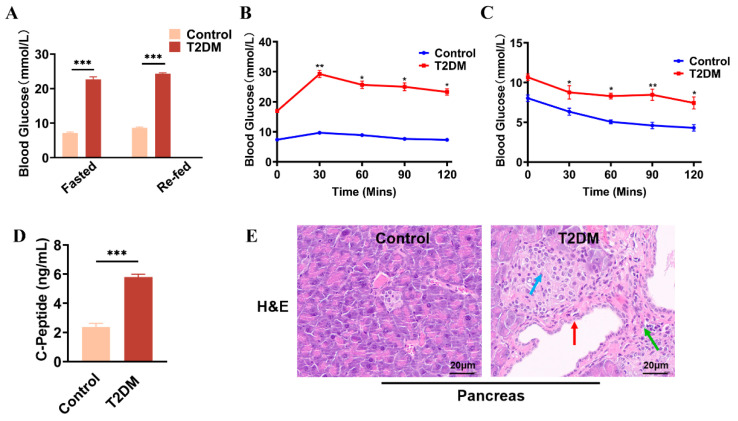
Characterization of HFD/STZ-induced T2DM mouse model. (**A**). Blood glucose levels measured from tail vein blood samples in fasted and refed mice were measured using a glucometer. (**B**). OGTT. (**C**). IPITT. (**D**). Serum C-peptide levels measured by ELISA. (**E**). H&E staining of pancreatic tissues was visualized under an optical microscope (scale bar = 20 µm) * *p* < 0.05, ** *p* < 0.01, *** *p* < 0.001.

**Figure 3 ijms-26-11469-f003:**
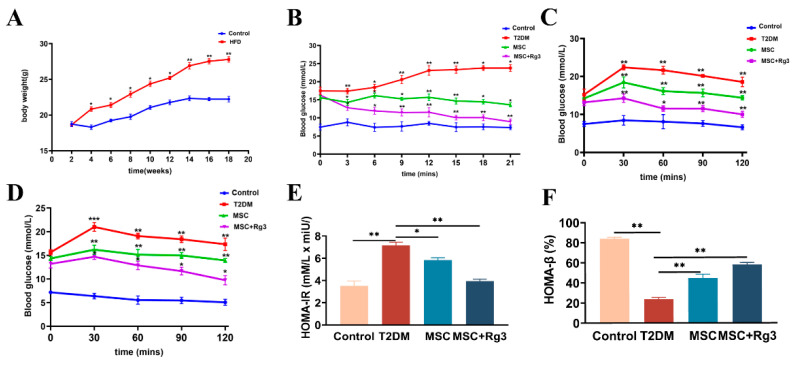
HUCMSCs Combined with 20(R)-Rg3 Improve Insulin Sensitivity in T2DM Mice. (**A**). The changes in body weight in the Control and HFD groups. (**B**). Blood glucose levels measured every three days. (**C**). Assesses individual glucose tolerance through an oral glucose tolerance test (OGTT). (**D**). Evaluates individual insulin resistance using an intraperitoneal insulin tolerance test (IPITT). (**E**). HOMA-IR. (**F**). HOMA-β. * *p* < 0.05, ** *p* < 0.01, *** *p* < 0.001.

**Figure 4 ijms-26-11469-f004:**
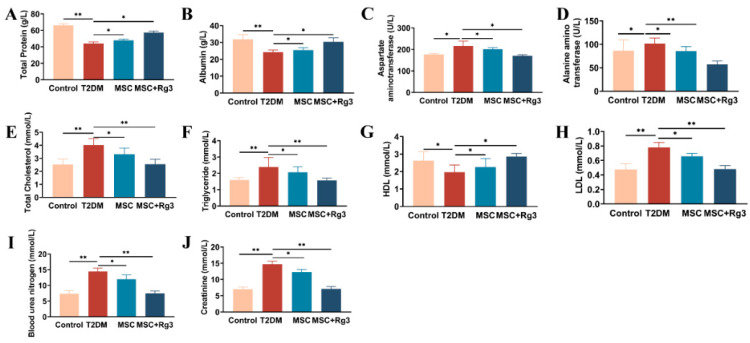
HUCMSCs Combined with 20(R)-Rg3 Improve Insulin Sensitivity in T2DM Mice. (**A**). TP. (**B**). ALB. (**C**). AST. (**D**). ALT. (**E**). TC. (**F**). TG. (**G**). HDL. (**H**). LDL. (**I**). BUN. (**J**). Cr. * *p* < 0.05, ** *p* < 0.01.

**Figure 5 ijms-26-11469-f005:**
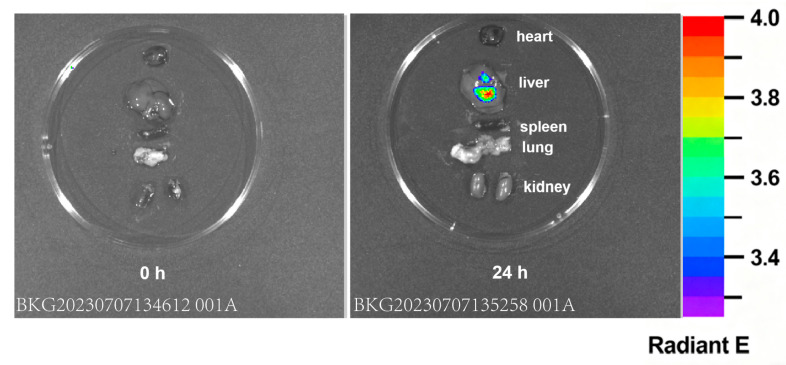
In vivo imaging showing organ distribution of transplanted cells.

**Figure 6 ijms-26-11469-f006:**
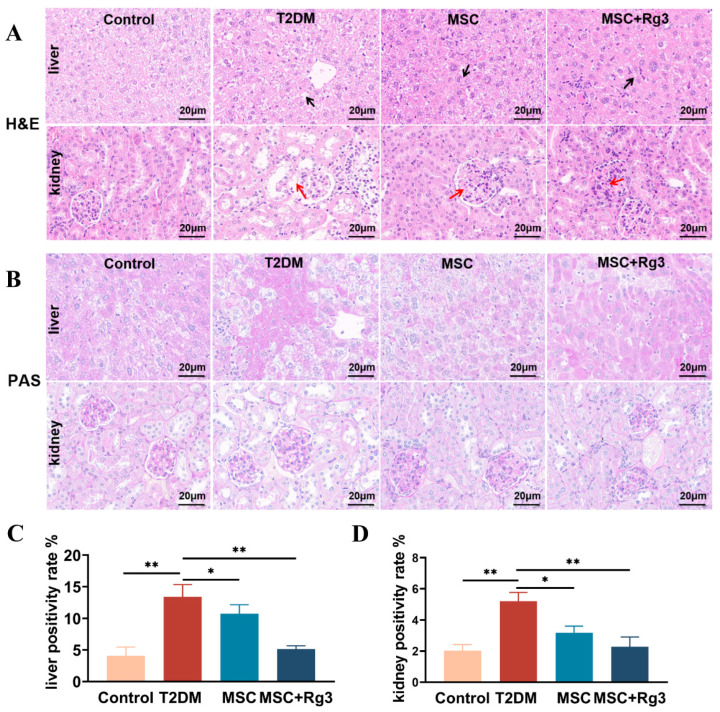
HUCMSCs combined with 20(R)-Rg3 promote glycogen storage in T2DM mice. (**A**). H&E staining of liver and kidney tissues. (**B**). PAS of liver and kidney (scale bar = 20 µm). (**C**). Quantitative analysis of PAS-positive staining in the liver. (**D**). Quantitative analysis of PAS-positive staining in the kidney. * *p* < 0.05, ** *p* < 0.01.

**Figure 7 ijms-26-11469-f007:**
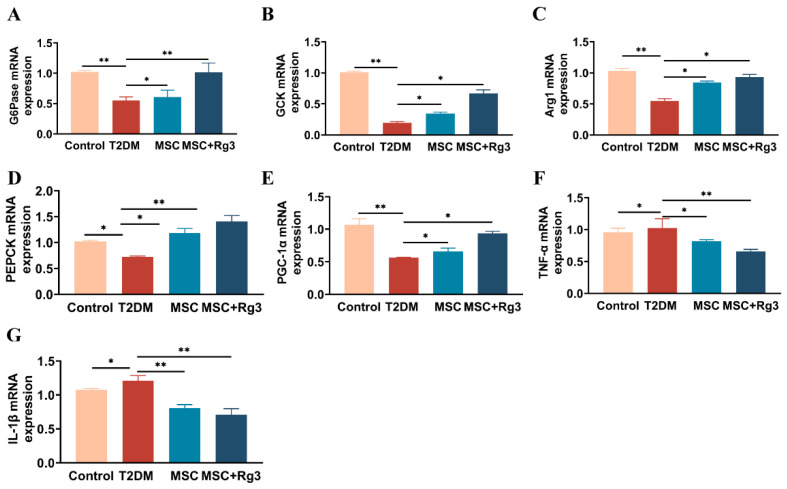
Effects of HUCMSCs Combined with 20(R)-Rg3 on Hepatic Gluconeogenic Enzymes and Inflammatory Markers. (**A**–**E**). mRNA expression levels of G6Pase, GCK, Arg-1, PEPCK, and PGC-1α. (**F**,**G**). mRNA expression levels of TNF-α and IL-1β. * *p* < 0.05, ** *p* < 0.01.

**Figure 8 ijms-26-11469-f008:**
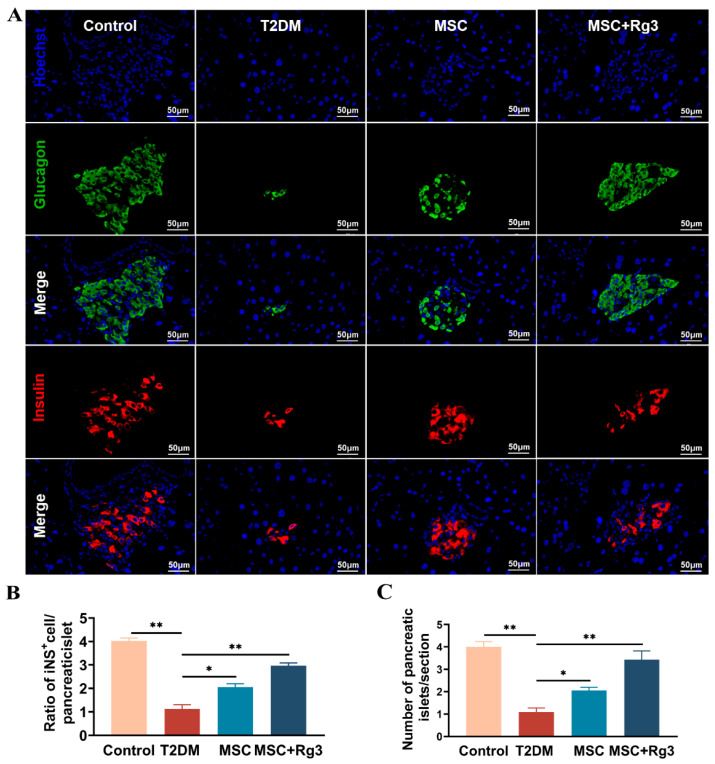
The combination of HUCMSCs and 20(R)-Rg3 enhances insulin secretion and promotes the “regeneration” of pancreatic islets. (**A**). Representative immunofluorescence images showing Hoechst (blue) Glucagon (green) and Insulin (red) staining (scale bar = 50 µm). (**B**). Quantitative analysis of islet numbers per pancreatic section. (**C**). Percentage of insulin-positive cells relative to total pancreatic areas. * *p* < 0.05, ** *p* < 0.01.

**Figure 9 ijms-26-11469-f009:**
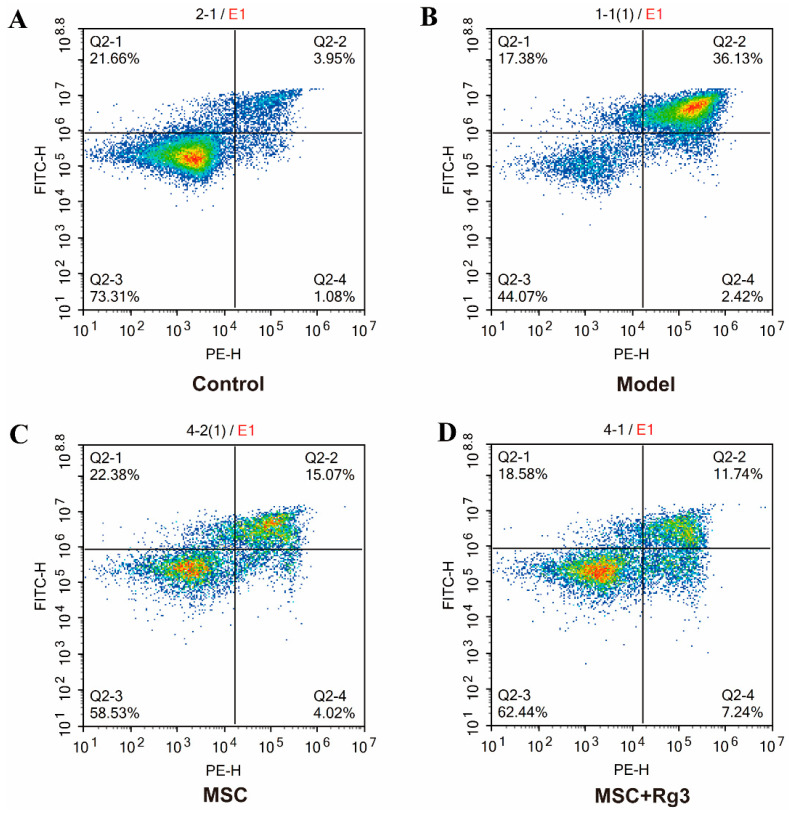
Representative flow-cytometric dot plots and quantitative apoptosis rates (Annexin V-FITC/PI). (**A**). Control group: Lowest apoptosis rate, predominantly viable cells (concentrated in the lower left quadrant). (**B**). T2DM group: Cells significantly increased in the upper right quadrants, with the highest apoptosis rate, indicating successful injury modeling. (**C**). HUCMSCs monotherapy group: The proportion of apoptotic cells decreased compared to the model group (**D**). HUCMSCs combined with 20(R)-Rg3 therapy group: Apoptotic cells further decreased.

**Figure 10 ijms-26-11469-f010:**
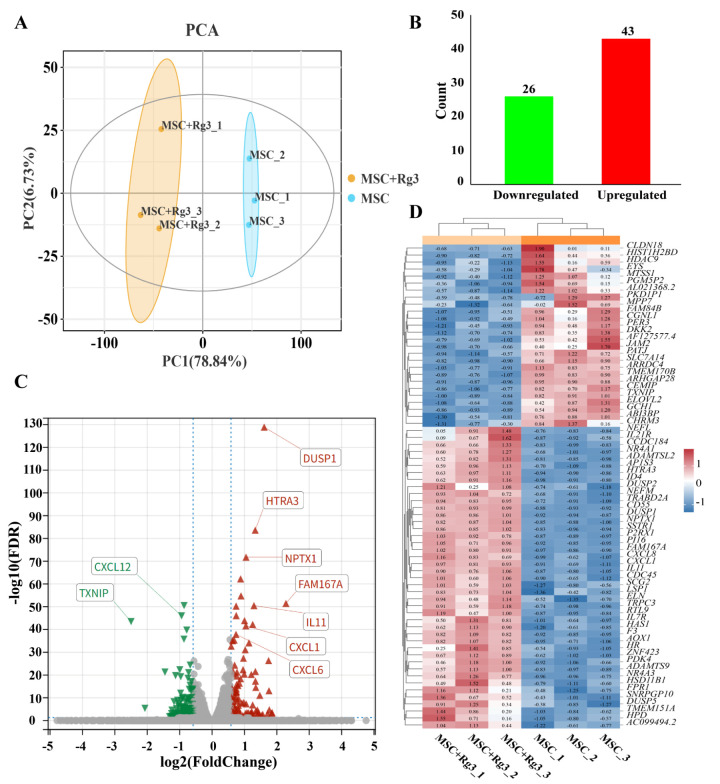
Transcriptomic profiling of HUCMSCs after 20(R)-Rg3 treatment. (**A**). PCA plot showing distinct clustering between 20(R)-Rg3-treated and control samples, with high consistency within biological replicates. (**B**). Bar plot showing the number of significantly upregulated and downregulated genes. (**C**). Volcano plot illustrates the distribution of differentially expressed genes; red dots indicate upregulated genes, green dots indicate downregulated genes, and gray dots indicate non-significant genes. (**D**). Heatmap of the top differentially expressed genes clustered across all samples.

**Figure 11 ijms-26-11469-f011:**
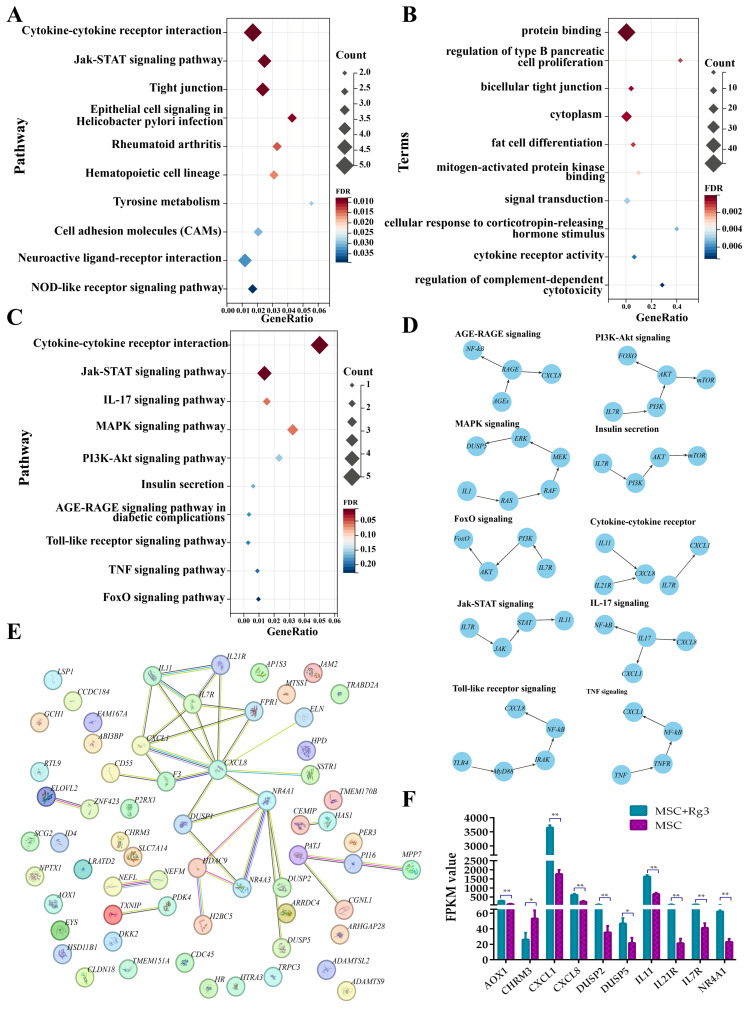
Enrichment and network analysis of DEGs in 20(R)-Rg3-treated HUCMSCs. (**A**). Bubble plot of the top 10 enriched GO terms across biological process, cellular component, and molecular function categories. (**B**) Bubble plot of the top 10 enriched KEGG pathways. (**C**). Bubble plot of signaling pathways directly or indirectly associated with diabetes. (**D**). Regulatory network of key genes in diabetes-related pathways. (**E**). PPI network of DEGs. (**F**). Expression patterns of DEGs enriched in diabetes-associated pathways. * *p* < 0.05, ** *p* < 0.01.

**Figure 12 ijms-26-11469-f012:**
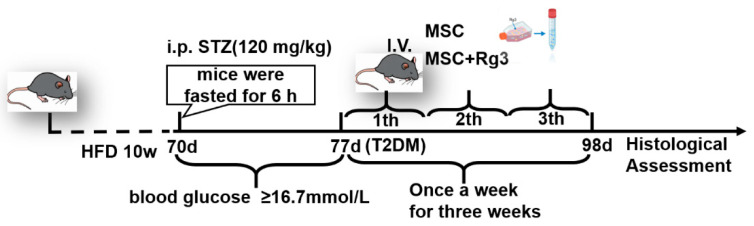
Schematic diagram of animal experimental design.

**Table 1 ijms-26-11469-t001:** Primer sequences in RT-qPCR.

Primer Name	Primer Sequence
GAPDH Forward Primer	GGTGAAGGTCGGTGTGAACG
GAPDH Reverse Primer	CTCGCTCCTGGAAGATGGTG
G6Pase Forward Primer	CGACTCGCTATCTCCAAGTGA
G6Pase Reverse Primer	GGGCGTTGTCCAAACAGAAT
PEPCK Forward Primer	CTGCATAACGGTCTGGACTTC
PEPCK Reverse Primer	GCCTTCCACGAACTTCCTCAC
GCk Forward Primer	AGGAGGCCAGTGTAAAGATGT
GCk Reverse Primer	CTCCCAGGTCTAAGGAGAGAAA
IL-1β Forward Primer	GAAATGCCACCTTTTGACAGTG
IL-1β Reverse Primer	TGGATGCTCTCATCAGGACAG
TNF-α Forward Primer	CAGGCGGTGCCTATGTCTC
TNF-α Reverse Primer	CGATCACCCCGAAGTTCAGTAG
Arg1 Forward Primer	CTCCAAGCCAAAGTCCTTAGAG
Arg1 Reverse Primer	GGAGCTGTCATTAGGGACATCA
PGC-1 Forward Primer	TATGGAGTGACATAGAGTGTGCT
PGC-1 Reverse Primer	GTCGCTACACCACTTCAATCC

## Data Availability

The original contributions presented in this study are included in the article/Appendix A. Further inquiries can be directed to the corresponding authors.

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
