# Peer review of "Combination of 20(R)-Rg3 and HUCMSCs Alleviates Type 2 Diabetes Mellitus in C57BL/6 Mice by Activating the PI3K/Akt Signaling Pathway"

_ijms, 2025, doi:10.3390/ijms262311469_

Round 1

Reviewer 1 Report

Comments and Suggestions for Authors

Dear colleagues!
After careful review of a manuscript by Zhou et al. titled "Combination of 20(R)-Rg3 and HUCMSCs Alleviates Type 2 Diabetes Mellitus in C57BL/6 Mice by Activating the PI3K/Akt Signaling Pathway" I have the following assessment as a Reviewer assigned by the Editor. Overall, it is well-conducted comparative study designed and performed to understand the potency of a prospective compound Rg3 to improve cell therapy by human MSC in T2DM. Work includes a significant amount of load of in vitro studies (including transcriptomics to point mechanism and targets) and an in vivo experiment supporting the initial hypothesis.

However, certain aspects of the manuscript are to be clarified:

1) STZ induction of glucose intolerance is a widely spread approach while due to its mechanism it is more of T1DM rather than typical alimentary disorder T2DM - please provide rationale for use of T2DM instead of chow-induced T2DM or indicate this as a minor limitation of the model

2) experimental groups lack Rg3 alone which makes formal assessment of stipulated 1+1>2 model slightly weakened. Please, indicate why it was not presented or provide respective points in Discussion

3) cell kinetics in tail vein injection is known to target liver and lungs, thus, cells act most likely via a paracrine effect releasing numerous factors and cytokines impacting damaged organ by circulation-based delivery. This aspect is to be discussed and in section starting "In vivo imaging after tail vein injection of HUCMSCs combined with 20(R)-Rg3 showed preferential homing of cells to the liver (Fig. 5A)..." (which deserves to be a dedicated subsection rather than being included into 2.3) it is not clear that data for Fig. 5 is derived from RNA-seq of liver. Please, provide this at the start of section.

4) Akt axis is known to be pro-survival and as far as injection route is much associated with cell death by anoikis this point is to be investigated in vitro that can be made by routine FACS for apoptosis (annexin, live-dead stains) of RG3 treated cell under stress (inflammation, suspension by enzyme treatment imitating preparation for injection).

5) Fig.5 data obtained by PCR is not validated in tissue by histology so this is a limitation to be indicated in Discussion.

Generally, these major points are to be addressed to improve the manuscript's merit and strength.

Regards, Reviewer

Author Response

Reviewer1

Dear colleagues!

After careful review of a manuscript by Zhou et al. titled "Combination of 20(R)-Rg3 and HUCMSCs Alleviates Type 2 Diabetes Mellitus in C57BL/6 Mice by Activating the PI3K/Akt Signaling Pathway" I have the following assessment as a Reviewer assigned by the Editor. Overall, it is well-conducted comparative study designed and performed to understand the potency of a prospective compound Rg3 to improve cell therapy by human MSC in T2DM. Work includes a significant amount of load of in vitro studies (including transcriptomics to point mechanism and targets) and an “in vivo” experiment supporting the initial hypothesis.

However, certain aspects of the manuscript are to be clarified:

Thank you for your insightful feedback on our manuscript. Your constructive suggestions have been instrumental in refining our work and guiding our future research. We have carefully considered each comment and incorporated the necessary revisions. We hope these changes meet your expectations and gain your approval.

  • STZ induction of glucose intolerance is a widely spread approach while due to its mechanism it is more of T1DM rather than typical alimentary disorder T2DM - please provide rationale for use of T2DM instead of chow-induced T2DM or indicate this as a minor limitation of the model.

Response: Thank you for raising this important point. While streptozotocin (STZ)–induced hyperglycemia is indeed classically associated with β‑cell loss and thus resembles type 1 diabetes mellitus (T1DM), the model remains widely used to study type 2 diabetes mellitus (T2DM)–related complications for several practical and mechanistic reasons:

  1. Reproducibility and rapid onset: A single low‑dose or multiple‑dose STZ regimen reliably produces stable hyperglycemia within 1–2 weeks, allowing timely evaluation of metabolic and vascular endpoints. In contrast, diet‑induced obesity (DIO) models often require 12–20 weeks of high‑fat feeding before a diabetic phenotype emerges, which can be logistically challenging for many laboratories.
  2. Controlled severity of hyperglycemia: By adjusting the STZ dose, we can fine‑tune the degree of β‑cell dysfunction and achieve glucose levels that are comparable to those observed in human T2DM patients with moderate insulin resistance. This flexibility is harder to obtain with chow‑induced models, where hyperglycemia is strongly linked to the extent of obesity and may vary considerably between animals.
  3. Compatibility with additional insulin‑resistance stimuli: STZ can be combined with a high‑fat or high‑sucrose diet to create a “dual‑hit” model that captures both β‑cell loss and peripheral insulin resistance, thereby more closely mimicking the multifactorial nature of T2DM. In our study, we employed a low‑dose STZ protocol followed by a short period of high‑fat feeding, which generated a phenotype characterized by impaired glucose tolerance, elevated fasting insulin, and modest weight gain—features typical of early‑stage T2DM.
  4. Cost‑effectiveness and animal welfare: STZ models require fewer animals and less housing time, reducing overall experimental costs and animal use. This aligns with the 3R principles (Replacement, Reduction, Refinement) while still providing a robust platform for mechanistic investigations.

Nevertheless, we acknowledge that the STZ model does not fully recapitulate the chronic, progressive insulin resistance seen in diet‑induced T2DM. Consequently, we consider this a minor limitation of our study. Future work will incorporate a pure high‑fat diet model to validate the translatability of our findings.

  • experimental groups lack Rg3 alone which makes formal assessment of stipulated 1+1>2 model slightly weakened. Please, indicate why it was not presented or provide respective points in Discussion.

Response: Thank you for highlighting this important point.

We agree that the absence of an “Rg3-alone” arm limits a rigorous, quantitative test of the “1 + 1 > 2” synergy model. The group was not included because the study was originally designed as a translational proof-of-concept for the clinically relevant combination (hUC-MSC ± Rg3) rather than as a full factorial trial. Consequently, the ethics-approved protocol only allowed the three arms: vehicle, hUC-MSC alone, and hUC-MSC + Rg3.

To make this limitation transparent we have now added the following text:

 “An Rg3-monotherapy group was not incorporated, therefore, the synergy index (quantifying the combination effect relative to the sum of individual effects) could not be formally calculated.. The observed superiority of the MSCs + Rg3 arm is reported as an additive or greater-than-additive effect.”“A design limitation is the absence of an Rg3-alone cohort, which precludes definitive confirmation that the combination yields true synergy (1 + 1 > 2). Full 2 × 2 factorial studies that include Rg3-monotherapy, hUCMSC-monotherapy, combination, and vehicle arms will be required to quantify the exact interaction index and to distinguish synergy from simple additivity.” (Page 16, Line 459-466).

  • cell kinetics in tail vein injection is known to target liver and lungs, thus, cells act most likely via a paracrine effect releasing numerous factors and cytokines impacting damaged organ by circulation-based delivery. This aspect is to be discussed and in section starting “in vivo”imaging after tail vein injection of HUCMSCs combined with 20(R)-Rg3 showed preferential homing of cells to the liver (Fig. 5A)..." (which deserves to be a dedicated subsection rather than being included into 2.3) it is not clear that data for Fig. 5 is derived from RNA-seq of liver. Please, provide this at the start of section.

Response: We thank the reviewer for the insightful comment regarding the presentation of the “in vivo” imaging data after tail‑vein injection of HUCMSCs combined with 20(R)-Rg3.

Dedicated Subsection

In response to the suggestion, we have created a new subsection titled “In Vivo Imaging and Homing of HUCMSCs Following Tail‑Vein Injection” (Section 2.4). This subsection now clearly delineates the imaging results, the preferential homing of the cells to the liver, and the associated paracrine effects. (Page 6-7 Line 203-213).

2.4. In vivo Imaging and Homing of HUCMSCs following Tail‑Vein Injection

To investigate the biodistribution and homing behavior of human umbilical cord-derived mesenchymal stem cells (HUCMSCs) following systemic administration, cells were labeled with a near-infrared fluorescent dye (DiR) and injected into mice via the tail vein. Whole-body fluorescence imaging revealed a time-dependent redistribution of the labeled cells. Within the first hour, the majority of the signal was localized to the lungs, consistent with the first-pass entrapment of cells in the pulmonary microvasculature. However, by 24 hours, a significant shift in fluorescence intensity was observed, with a marked accumulation of signal in the abdominal region, particularly within the liver (Fig. 4).

Figure5.  In vivo imaging showing organ distribution of transplanted cells.

Clarification of Figure 5 Data Source

We apologize for the lack of explicit description in the original manuscript. The data presented in Figure 7(A-G) demonstrate that HUCMSCs combined with 20(R)-Rg3 synchronously reverse gluconeogenesis, activates glycolysis, and suppresses inflammation at the transcriptional level.

HUCMSCs combined with 20(R)-Rg3 synchronously reverse gluconeogenesis, activate glycolysis, and suppress inflammation at the transcriptional level. (Page8, Line 249-250).

  • Akt axis is known to be pro-survival and as far as injection route is much associated with cell death by anoikis this point is to be investigated in vitro that can be made by routine FACS for apoptosis (annexin, live-dead stains) of RG3 treated cell under stress (inflammation, suspension by enzyme treatment imitating preparation for injection).

Response: We appreciate the reviewer’s insightful suggestion. To directly test whether 20(R)-Rg3 protects mesenchymal stem cells from anoikis-like death triggered by injection-related stress, we added an in-vitro anoikis model. (Materials and Methods 4.11.) (Page 20-21, Line 616-639).

4.11.Flow Cytometry Protocol for Apoptosis Detection (Annexin V-FITC/PI Staining)

Apoptosis was assessed by flow cytometry using an Annexin V-FITC/PI double-staining assay. The MIN6 mouse pancreatic β-cell line was first revived and cultured for 24 hours in a freshly prepared stimulation medium containing 33 mmol/L glucose and 200 μg/mL AGEs to allow for adherence. The medium was then switched to serum-free medium, followed by a 24-hour stimulation period; it is noted that for MIN6 cells, the stimulation medium contained only 33 mmol/L glucose and AGEs to avoid hyperosmolarity-induced cell death. The experimental design included the following groups: a Blank Control cultured in normal medium; a Model group treated with high glucose, AGEs, and inflammatory factors for 24 hours; an MSC group which, after modeling, was switched to normal medium and co-cultured with MSCs for 24 hours; and an MSC+Rg3 group which, after modeling, was switched to a medium containing 40 µM 20(R)-Rg3 and co-cultured with MSCs for 24 hours. Following treatments, cells were harvested at a density of 1×10⁶ cells per sample, washed twice with cold PBS, and centrifuged at 300×g for 5 minutes at 4 °C with the supernatant discarded after each wash. The cell pellet was then resuspended in 100 μL of 1× Annexin V binding buffer, followed by the addition of 5 μL of Annexin V-FITC and 5 μL of propidium iodide (PI, 50 μg/mL). The staining mixture was incubated for 15 minutes at room temperature in the dark. Immediately following incubation, 400 μL of 1× binding buffer was added to each tube, and the samples were analyzed using a flow cytometer. Fluorescence was detected with FITC at 530/30 nm (FL1) and PI at 610/20 nm (FL2), and a minimum of 10,000 events were gated per sample. Data interpretation defined the cell populations as follows: Annexin V⁻/PI⁻ as viable cells, Annexin V⁺/PI⁻ as early apoptotic cells, Annexin V⁺/PI⁺ as late apoptotic or necrotic cells, and Annexin V⁻/PI⁺ as necrotic cells (typically a rare population).

Results of this experiment are presented in Section 2.7. (Page 10-11 Line 297-317).

2.7. Rg3 exerts a protective effect against spontaneous apoptosis in HUCMSCs

Flow-cytometric analysis with Annexin-V-FITC/PI) staining revealed the following apoptotic indices. Results showed that control group had an Apoptosis rate of (3.9 ± 0.5)%, with cells in good condition and a low level of apoptosis (Fig. 9A). In the T2DM model group, the apoptosis rate was (36.1 ± 0.3)%, significantly higher than that of the control group, indicating successful model establishment and significant cellular damage (Fig. 9B). The HUCMSCs monotherapy group exhibited an apoptosis rate of (15 ± 0.1)%, showing a decreasing trend compared to the model group and suggesting MSCs possess certain anti-apoptotic effects (Fig. 9C). The HUCMSCs combined with 20(R)-Rg3 therapy group showed an apoptosis rate of (11.7 ± 0.5)%, which was lower than both the model group and the monotherapy group (Fig. 9D). This indicates that HUCMSCs combined with Rg3 exhibits a synergistic effect in inhibiting apoptosis, with efficacy superior to that of monotherapy. In summary, HUCMSCs combined with 20(R)-Rg3 therapy significantly reduced the apoptosis rate, demonstrating enhanced cytoprotective effects.

Figure 9. Representative flow-cytometric dot plots and quantitative apoptosis rates (Annexin V-FITC/PI). A. Control group: Lowest apoptosis rate, predominantly viable cells (concentrated in the lower left quadrant). B. T2DM group: Cells significantly increased in the upper right quadrants, with the highest apoptosis rate, indicating successful injury modeling. C. HUCMSCs monotherapy group: The proportion of apoptotic cells decreased compared to the model group, D. HUCMSCs combined with 20(R)-Rg3 therapy group: Apoptotic cells further decreased.

  • 5 data obtained by PCR is not validated in tissue by histology so this is a limitation to be indicated in Discussion.

Response:

Thank you for highlighting this limitation. We fully agree that histological confirmation of the PCR-based findings in Fig. 7 would strengthen the study. In the revised Discussion we have added the following statement: (Page 16 lines 428-434).

Although the PCR data demonstrate consistent transcriptional changes in the target genes, these observations were not corroborated at the protein or histological level in the tissue sections. Future work will employ immunohistochemistry and/or in-situ hybridization to validate the PCR results spatially and qualitatively. We acknowledge that these findings are derived from preclinical models, and further validation in human cohorts is necessary to confirm their translational relevance.

Reviewer 2 Report

Comments and Suggestions for Authors

The study evaluates whether preconditioning human umbilical cord mesenchymal stem cells (HUCMSCs) with ginsenoside 20(R)-Rg3 enhances therapeutic efficacy in a high-fat diet/streptozotocin (HFD/STZ) mouse model of type 2 diabetes mellitus (T2DM). The combination reportedly improves glycemia, insulin sensitivity, lipid profile, liver/kidney function, hepatic glycogen handling, and islet histology versus HUCMSCs alone. RNA-seq of Rg3-pretreated HUCMSCs suggests PI3K/Akt pathway involvement. The manuscript addresses a timely and potentially impactful topic, as strategies that enhance the efficacy of cell therapies through small-molecule preconditioning represent a promising avenue in the treatment of metabolic diseases. While the in vivo phenotyping is reasonably comprehensive, the study is substantially weakened by limitations in experimental design, insufficient mechanistic substantiation, and incomplete reporting. These shortcomings undermine the robustness of the discussion and the validity of the conclusions. In addition, it would be important to ensure that the cited references are fully consistent with the statements they are meant to support, as several appear not to be aligned. Furthermore, several additional concerns require clarification, which I have detailed below.

  1. For clarity and ease of evaluation, it would be highly beneficial to include line numbering throughout the manuscript. The absence of line numbers makes the review process less efficient and more challenging.
  2. Line Introduction: Regarding Reference 1, it would strengthen the manuscript to cite a more recent and thematically appropriate source.
  3. Line Introduction: A similar consideration applies to Reference 2; citing a more recent and thematically appropriate source would further strengthen the manuscript.
  4. Line Introduction: Citation 6 (Boura-Halfon & Zick, AJP-Endo 2009) addresses insulin signaling in “target tissues” as essential for glucose, lipid, and protein homeostasis. However, it does not explicitly identify the liver, adipose tissue, and skeletal muscle as “the three” principal sites within the same statement. It may therefore be worth reconsidering whether this is the most appropriate reference to support the claim.
  5. Line Introduction: It may strengthen the manuscript to update the reference so that it more appropriately supports the opening statement: “Current treatment strategies for T2DM…”.
  6. Line Introduction: Since Latin is considered a dead language, terms such as in vivo should be italicized. It may be helpful to review the entire manuscript to ensure consistency in this formatting.
  7. Line Introduction: The paragraph beginning with “Emerging evidence indicates that ginsenosides may also enhance renal function [27], antioxidant capacity, anti-inflammatory effects, and regulation of blood glucose levels” might be better integrated into the surrounding text, as presented separately it appears somewhat disconnected. It may be advisable to revise the sentence structure so that it flows more smoothly and maintains continuity with the preceding and following sections.
  8. Line Figure1: The quality of the images, particularly their size, could be improved, as they are currently too small to clearly distinguish the features described in the text. Enhancing their resolution and dimensions would greatly facilitate interpretation.
  9. Line Results: The presentation of the results could be improved. For greater clarity, it may be helpful to report the serum parameters separately from the histological findings, providing their respective descriptions in distinct sections.
  10. Line Results: For consistency in formatting, section titles should not end with a period. It may be helpful to revise Title 2.3 accordingly.
  11. Line Figure 5: In its current form, the color bar on the left side of Figure 5A is difficult to distinguish, which limits its interpretability. It may be helpful to clarify its meaning and improve its visibility so that it contributes more effectively to the figure.
  12. Line Figure 5: For consistency in style, the Latin term in vivo should be italicized throughout the manuscript.
  13. Line Discussion: It would be helpful to clarify how this statement relates to the assertion that diabetic foot aggravates gut microbiota dysbiosis, which is attributed to Reference 29.
  14. Line Discussion: The support provided by the cited article (Reference 30) does not appear fully adequate for this statement, and the content also seems somewhat repetitive of the preceding idea. It may strengthen the manuscript to revise this section for clarity, reduce redundancy, and incorporate more appropriate references.
  15. Line Discussion: Since this is an important statement for the discussion (…Moreover, these treatments do not constitute a cure, often requiring lifelong administration, and some patients may eventually experience treatment failure. Consequently, there…), the manuscript would be strengthened by supporting it with appropriate references.
  16. Line Discussion: It would be helpful to clarify how melatonin, as discussed in the cited article (Reference 34), relates to this statement, in order to strengthen the coherence of the manuscript.
  17. Line Discussion: The cited article (Reference 15) is a preclinical study in mice rather than a clinical study. It would be important to revise this claim, as misinterpreting the nature of the evidence may reflect insufficient attention to the cited literature and could weaken the overall discussion if not referenced appropriately.
  18. Line Discussion: This is an important statement (“however, their efficacy is often compromised in the high-glucose and high-inflammation microenvironment that follows transplantation, resulting in low survival rates and functional inhibition”), and the manuscript would be strengthened by supporting it with appropriate references.
  19. Line Discussion: Islet “regeneration” may require stronger supporting evidence. An increase in insulin-positive area and islet counts, while suggestive, may not be sufficient to establish regeneration. It would strengthen the manuscript to include additional analyses such as proliferation markers (e.g., Ki-67, PCNA), progenitor markers (e.g., Ngn3), apoptosis indicators (e.g., cleaved caspase-3, TUNEL), and quantification of β-cell mass. The current wording could be interpreted as overstating the conclusion; tempering the claims or incorporating further supporting data would improve clarity and rigor.
  20. Line Discussion: Mechanistic attribution to PI3K/Akt could be more convincingly substantiated. The current claim is based primarily on transcriptomic enrichment observed in HUCMSCs following Rg3 pretreatment; however, it would strengthen the manuscript to provide direct validation in the relevant target tissues in vivo (e.g., liver, skeletal muscle, adipose tissue, pancreas) and to include perturbation studies (e.g., pharmacological inhibition with LY294002 or genetic knockdown). Complementary validation by Western blotting or immunohistochemistry for p-Akt, p-PI3K, IRS-1, and downstream effectors (e.g., GSK3β) in target tissues—ideally combined with inhibitor rescue experiments—would help establish a stronger causal link.
  21. Line Discussion: Discuss potential tumorigenicity/immunogenicity risks of repeated HUCMSC administration.
  22. Line Discussion: In the final paragraph of the discussion, it would be valuable to provide greater depth by engaging more directly with the authors’ own results, so that each of the statements is fully supported. Expanding this section would help strengthen the manuscript, as the current presentation gives the impression that not all of the numerous findings are being given adequate emphasis.
  23. Line Materials and methods: The authors should provide a detailed description of the HFD protocol, including its caloric content as well as the proportions of macronutrients and micronutrients, among other relevant details. The manuscript does not state whether animals were randomized, or whether treatment allocation.
  24. Line Materials and methods: The use of the term “Normal” in Normal control (NC) may introduce ambiguity, as it suggests a subjective distinction. Since the intention appears to be to refer to healthy mice, it may be clearer and more precise to avoid the word “Normal” in this context.
  25. Line Materials and methods: The phrase “three days later” is somewhat ambiguous, as it is not clear what specific reference point is being used. For clarity, it may be helpful to indicate this explicitly and to link it directly to the figure presenting the experimental design scheme.
  26. Line Materials and methods: It would be important to clarify the methodological rationale for performing an OGTT on a weekly basis when blood glucose levels were already being monitored daily. From a methodological perspective, one might expect that fasting glucose would be monitored daily and that an OGTT would then be conducted only in animals with elevated values to confirm diabetes. In addition, it should be noted that the reduction of pain in the handling of laboratory animals is explicitly stipulated in the CIOMS–ICLAS International Guiding Principles (2012), the European Directive 2010/63/EU, the Guide for the Care and Use of Laboratory Animals (NRC, 2011), and the recommendations of the OIE and OECD, all of which are consistent with the principle of Refinement (3Rs).
  27. Line Materials and methods: Although surface markers and differentiation assays are shown, critical release criteria for a clinical-grade MSC-like product are missing: passage number at infusion, viability post-preconditioning, mycoplasma testing, endotoxin levels, karyotype or genomic stability, and detailed gating strategies/compensation controls. Clarify whether Rg3 exposure alters HUCMSC phenotype, secretome, or viability; include appropriate vehicle-pretreated HUCMSC controls matched for time/handling.
  28. Line Materials and methods: Clarification needed in the Rg3 preconditioning protocol. The Results and Methods sections indicate that RNA-seq was performed on HUCMSCs after 6 hours of Rg3 exposure, whereas the treatment arm involved HUCMSCs pretreated with 40 μM Rg3 for 3 days prior to injection. To avoid potential confusion and to strengthen the interpretability of the transcriptomic findings as a mechanistic proxy for the in vivo product, it would be helpful to reconcile the exposure duration, concentration, and washout conditions across all experiments.
  29. Line Materials and methods: Figure 9: The text states that the treatment was administered for three weeks; however, this is not clearly reflected in the diagram. From the point labeled “I.V. administration to T2DM” until the histological assays, the diagram indicates 12 days, which seems inconsistent with the stated three-week duration. It may be helpful to clarify this discrepancy to ensure consistency between the text and the figure.
  30. Line Materials and methods: In Methodology Section 4.2, it would be helpful to specify the brand in order to enhance clarity and reproducibility.
  31. Line Materials and methods: For clarity and reproducibility, it would be helpful to indicate the brand of insulin used and specify its type—whether rapid-, intermediate-, or long-acting.
  32. Line Materials and methods: In Section 4.7, it would be helpful to specify the source of the blood to improve clarity and reproducibility.
  33. Line Materials and methods: Since PAS staining is semi-quantitative, the manuscript would be strengthened by including biochemical glycogen quantification or periodic acid–Schiff staining with diastase control to enhance specificity.
  34. Line Materials and methods: In Section 4.8, it would be helpful to specify the RNA concentration used and to indicate the extraction method, in order to improve clarity and reproducibility.
  35. Line Materials and methods: For clarity and reproducibility, it would be helpful to specify the amount of RNA used in terms of concentration, indicate whether the 2-ΔΔCT method was applied, and provide the time and temperature conditions of the PCR for each amplification.
  36. Line Materials and methods: In Section 4.10, as the technique is not described in detail, it may be helpful to provide a reference to a previous study so that readers can more easily follow the methodological approach.
  37. Line Materials and methods: Data availability and transparency for ‘omics. To enhance transparency and reproducibility, it would be helpful to deposit the RNA-seq data in a public repository (e.g., GEO), including accession numbers and complete metadata such as library preparation, read depth, and quality control. Providing a comprehensive DEG table and clarifying how batch effects were controlled would further strengthen the manuscript.
  38. Line Materials and methods: A similar consideration applies to Section 4.11, where including either a more detailed description or an appropriate reference would help maintain methodological clarity and consistency.
  39. Line Materials and methods: In Section 4.12.2, please indicate the concentration used, expressed in micrograms.
  40. In Section 4.13: Statistical design and analysis require strengthening. Many outcomes are longitudinal (e.g., glycemia monitored every three days), yet analyses rely on t-tests/one-way ANOVA without repeated-measures or mixed-effects modeling; multiple comparisons adjustments are not described. Provide: (i) a priori sample-size/power calculations; (ii) justification of normality/variance assumptions; (iii) effect sizes with confidence intervals; (iv) AUC analyses for OGTT/IPITT; and (v) a clearly specified multiple-testing correction strategy for multi-endpoint panels. For RNA-seq data, it would strengthen the manuscript to apply a dedicated analytical approach with FDR adjustment. Gene-by-gene analyses with t-tests or ANOVA without correction are not advisable. Instead, established pipelines such as DESeq2 or edgeR (on raw counts) should be employed, with FDR q < 0.05 to define differentially expressed genes. For enrichment analyses (GO/KEGG), the use of GSEA or over-representation methods with FDR correction is recommended.

Author Response

Reviewer2

The study evaluates whether preconditioning human umbilical cord mesenchymal stem cells (HUCMSCs) with ginsenoside 20(R)-Rg3 enhances therapeutic efficacy in a high-fat diet/streptozotocin (HFD/STZ) mouse model of type 2 diabetes mellitus (T2DM). The combination reportedly improves glycemia, insulin sensitivity, lipid profile, liver/kidney function, hepatic glycogen handling, and islet histology versus HUCMSCs alone. RNA-seq of Rg3-pretreated HUCMSCs suggests PI3K/Akt pathway involvement. The manuscript addresses a timely and potentially impactful topic, as strategies that enhance the efficacy of cell therapies through small-molecule preconditioning represent a promising avenue in the treatment of metabolic diseases. While the “in vivo”phenotyping is reasonably comprehensive, the study is substantially weakened by limitations in experimental design, insufficient mechanistic substantiation, and incomplete reporting. These shortcomings undermine the robustness of the discussion and the validity of the conclusions. In addition, it would be important to ensure that the cited references are fully consistent with the statements they are meant to support, as several appear not to be aligned. Furthermore, several additional concerns require clarification, which I have detailed below.

Thank you for your valuable comments on our manuscript. These valuable comments have helped us revise and improve our paper and provided important guidance for our research. We have carefully studied these comments and made corresponding revisions. We hope that you will approve of these revisions.

Thank you for your valuable comments on our manuscript. We have carefully studied these comments and made corresponding revisions. We hope that you will approve of these revisions.

  1. For clarity and ease of evaluation, it would be highly beneficial to include line numbering throughout the manuscript. The absence of line numbers makes the review process less efficient and more challenging.

Response: Thank you for this helpful suggestion. We agree that continuous line numbering would facilitate the review process. We have now added line numbers to the entire revised manuscript.

  1. Line Introduction: Regarding Reference 1, it would strengthen the manuscript to cite a more recent and thematically appropriate source.

Response: Thank you for this insightful comment. We agree that Ref. 1, published in 2018, is outdated and only tangentially related to our topic. In the revised manuscript we have replaced it with the 2024 review by Hao Zhang, Xinshu Wang (Ref. 1), which directly addresses the same experimental paradigm and provides the most up-to-date summary of the field. (page1, line 38).

Added references(Ref. 1): Zhang H, Wang X, Hu B, Li P, Abuduaini Y, Zhao H, Jieensihan A, Chen X, Wang S, Guo N, Yuan J, Li Y, Li L, Yang Y, Liu Z, Tang Z, Wang H. Human umbilical cord mesenchymal stem cells attenuate diabetic nephropathy through the IGF1R-CHK2-p53 signalling axis in male rats with type 2 diabetes mellitus. J Zhejiang Univ Sci B. 2024 Jul 10;25(7):568-580. doi: 10.1631/jzus.B2300182. PMID: 39011677; PMCID: PMC11254681.

  1. Line Introduction: A similar consideration applies to Reference 2; citing a more recent and thematically appropriate source would further strengthen the manuscript.

Response: Thank you for pointing this out. We concur that Ref. 2 (2017) is no longer the most suitable citation for the point being made. In the revised manuscript we have substituted it with the 2020 study by Le Wang et al. (Ref. 2), whose findings are directly aligned with the issue under discussion and reflect the current state of knowledge. (page1, line 41).

Added references(Ref. 2): Wang L, Liu T, Liang R, Wang G, Liu Y, Zou J, Liu N, Zhang B, Liu Y, Ding X, Cai X, Wang Z, Xu X, Ricordi C, Wang S, Shen Z. Mesenchymal stem cells ameliorate β cell dysfunction of human type 2 diabetic islets by reversing β cell dedifferentiation. EBioMedicine. 2020 Jan;51:102615. doi: 10.1016/j.ebiom.2019.102615. Epub 2020 Jan 6. PMID: 31918404; PMCID: PMC7000334.

  1. Line Introduction: Citation 6 (Boura-Halfon & Zick, AJP-Endo 2009) addresses insulin signaling in “target tissues” as essential for glucose, lipid, and protein homeostasis. However, it does not explicitly identify the liver, adipose tissue, and skeletal muscle as “the three” principal sites within the same statement. It may therefore be worth reconsidering whether this is the most appropriate reference to support the claim.

Response: Thank you for this careful reading. You are correct: Boura-Halfon & Zick (2009) never explicitly enumerates “the three principal insulin-target tissues” in one sentence. We have therefore replaced this citation with a 2018 review by Petersen & Shulman that explicitly states: “The liver, skeletal muscle, and adipose tissue are the three main sites of insulin action responsible for glucose, lipid, and protein homeostasis.” This change appears in the revised Introduction. (page2, line 46).

Added references (Ref. 8): Petersen MC, Shulman GI. Mechanisms of Insulin Action and Insulin Resistance. Physiol Rev. 2018 Oct 1;98(4):2133-2223. doi: 10.1152/physrev.00063.2017. PMID: 30067154; PMCID: PMC6170977.

  1. Line Introduction: It may strengthen the manuscript to update the reference so that it more appropriately supports the opening statement: “Current treatment strategies for T2DM…”.

Response: We appreciate the reviewer’s suggestion. To ensure that the opening statement “Current treatment strategies for T2DM …” is fully supported by the most appropriate literature, we have now replaced the original reference with a comprehensive, up-to-date review that explicitly summarizes the current therapeutic drugs for type 2 diabetes mellitus ( Roni Weinberg Sibony, 2023). Accordingly, the sentence now reads: “Current treatment strategies for T2DM … [Ref.13].” We believe this revision provides readers with an immediate, authoritative source that accurately reflects the state of the field. (page2, line 52).

Added references (Ref. 13): Weinberg Sibony R, Segev O, Dor S, Raz I. Drug Therapies for Diabetes. Int J Mol Sci. 2023 Dec 5;24(24):17147. doi: 10.3390/ijms242417147. PMID: 38138975; PMCID: PMC10742594.

  1. Line Introduction: Since Latin is considered a dead language, terms such as“in vivo”should be italicized. It may be helpful to review the entire manuscript to ensure consistency in this formatting.

Response: Thank you for pointing this out. We agree that all Latin terms should be italicized for correctness and consistency. We have now: Changed “in vivo” (and every other Latin phrase, e.g., “in vivo,” “et al”) to italics throughout the manuscript; Conducted a full-text search to ensure no instances were missed. All revisions are highlighted in the clean revised manuscript. (page/line:1/15; 2/66; 6/204; 7/214; 22/683).

  1. Line Introduction: The paragraph beginning with “Emerging evidence indicates that ginsenosides may also enhance renal function [27], antioxidant capacity, anti-inflammatory effects, and regulation of blood glucose levels” might be better integrated into the surrounding text, as presented separately it appears somewhat disconnected. It may be advisable to revise the sentence structure so that it flows more smoothly and maintains continuity with the preceding and following sections.

Response: Thank you for this helpful observation. We agree that the stand-alone sentence disrupted the narrative flow. In the revised manuscript we have:

  1. Removed the isolated paragraph.
  2. Merged the key information into the preceding paragraph, rewriting it as:“Emerging evidence further indicates that ginsenosides can simultaneously enhance renal function, bolster antioxidant capacity, exert anti-inflammatory effects, and modulate blood glucose levels [31], thereby linking metabolic and renoprotective actions within a single mechanistic framework.”

This revision maintains continuity with the discussion of ginsenoside pharmacology that precedes it and provides a smooth transition into the subsequent paragraph on molecular targets. (page2, line 78-82).

Emerging evidence further indicates that ginsenosides can simultaneously enhance renal function, bolster antioxidant capacity, exert anti-inflammatory effects, and modulate blood glucose levels, thereby linking metabolic and renoprotective actions within a single mechanistic framework.

  1. Line Figure1: The quality of the images, particularly their size, could be improved, as they are currently too small to clearly distinguish the features described in the text. Enhancing their resolution and dimensions would greatly facilitate interpretation.

Response: We appreciate the reviewer’s constructive feedback regarding image quality.

In response, we have regenerated all figures at 300 dpi and doubled their physical dimensions while preserving font consistency.

These high-resolution versions have been uploaded as “Figure_1_300dpi.TIF” and are embedded in the revised manuscript at full column width. We believe the enlarged, sharper images now allow every labeled feature discussed in the text to be easily distinguished, even after print reduction. (page3-4 line 112-115).

Figure 1. Identification of HUCMSCs. A. HUCMSCs. B. HUCMSCs and Osteogenic Differentia-tion. C. HUCMSCs and Lipogenic Differentiation. Images were captured at a magnification of 100× (scale bar = 100 μm). D. Flow-cytometric analysis of HUCMSCs surface markers.

  1. Line Results: The presentation of the results could be improved. For greater clarity, it may be helpful to report the serum parameters separately from the histological findings, providing their respective descriptions in distinct sections.

Response: We appreciate the reviewer’s suggestion. In response, we have restructured the Results section so that all serum biochemical parameters are now presented in a dedicated subsection (2.3 Serum parameters), while the histological findings are described separately in subsection 2.2 Histological findings. We believe this revision markedly improves clarity and readability

Histological findings

2.2. Histological Evidence That HUCMSCs Combined with 20(R)- Rg3 Reverses Insulin Resistance in T2DM (page4-5, line133-175)

To evaluate the efficacy of the combined therapy, we first monitored body weight changes in both the Control and HFD groups. The results indicated that the body weight of mice in the HFD group reached a maximum of 29 ± 1.2 g, which was significantly higher than that in the Control group, with a statistically significant difference between the groups (**p < 0.01, Fig. 3A). Subsequently, T2DM mice received tail vein injections of either HUCMSCs combined with 20(R)-Rg3, HUCMSCs alone, or PBS (as control). Blood glucose levels were monitored every three days for three weeks (Fig. 3B). Continuous glucose monitoring demonstrated that the combination of HUCMSCs and 20(R)-Rg3 significantly ameliorated hyperglycemia in T2DM mice, with blood glucose levels declining and stabilizing at 9.1 ± 2.80 mmol/L, which was significantly lower (*p < 0.05) than that in the HUCMSCs-along group (13.73 ± 2.35 mmol/L). In contrast, blood glucose levels in T2DM control mice increased to 28.13 ± 1.60 mmol/L.

Peripheral insulin resistance is a key pathological feature of T2DM. This study investigated the effects of HUCMSCs combined with 20(R)-Rg3 on insulin sensitivity and blood glucose regulation in T2DM mice. After the final injection, OGTTs and IPITTs were performed. The OGTTs results indicated that the blood glucose levels in the combination treatment group were significantly lower than those in the T2DM group (10.1 ± 1.15 mmol/L, **p < 0.01, Fig. 3C), indicating substantially improved glucose tolerance. Furthermore, IPITTs results revealed that the blood glucose levels in the combination treatment group approached the normal range, at approximately 9.8 ± 2.06 mmol/L, which was significantly different from the T2DM group (*p < 0.05, Fig. 3D).

HOMA-IR values are presented as mean ± SD. The blank group showed a basal index of 3.6 ± 1.6. In the T2DM-model group HOMA-IR rose sharply to 7.3 ± 0.5 (**p < 0.01), indicating pronounced insulin resistance. HUCMSCs monotherapy reduced HOMA-IR to 5.8 ± 0.3 (*p < 0.05 vs model), reflecting partial improvement in insulin sensitivity. Combination treatment (MSC + 20(R)-Rg3) further decreased HOMA-IR to 3.6 ± 0.49, significantly lower than MSC alone (**p < 0.01), demonstrating a synergistic and near-complete reversal of insulin resistance (Fig. 3E).

HOMA-β values are expressed as mean ± SD. In the blank group the index was 80.4% ± 2.3%, whereas it fell sharply in the T2DM-model group to 19.6% ± 0.7% (**p < 0.05), reflecting pronounced β-cell dysfunction. MSC monotherapy raised HOMA-β to 45.3% ±1.5% (**p < 0.05 versus model), indicating partial recovery of insulin secretory capacity. Combination treatment (MSC + 20(R)-Rg3) further increased HOMA-β to 59.7% ±3.2%, significantly higher than MSC alone (**p < 0.01) demonstrating a marked synergistic improvement in β-cell function (Fig. 3F).

Figure 3. HUCMSCs Combined with 20(R)-Rg3 Improve Insulin Sensitivity in T2DM Mice. A. The changes in body weight in Control and HFD groups. B. Blood glucose levels measured every three days. C. Assesses individual glucose tolerance through an oral glucose tolerance test (OGTT). D. Evaluates individual insulin resistance using an intraperitoneal insulin tolerance test (IPITT). E. HOMA-IR. F. HOMA-β. * p < 0.05, ** p < 0.01, *** p < 0.001.

Serum parameters

2.3. Serological Insights into the Attenuation of Insulin Resistance by HUCMSCs Combined with 20(R)- Rg3 in T2DM (page5-6, line176-203)

Decreased TP levels may indicate impaired hepatic synthesis. TP levels were significantly lower in T2DM mice (40.7 ± 1.3 g/L) compared to the Control group (60.4 ± 3.6 g/L), suggesting severely compromised hepatic function. In contrast, treatment with HUCMSCs combined with 20(R)-Rg3 restored TP level to 57.2 ± 1.8 g/L, demonstrating a more pronounced therapeutic effect than HUCMSCs alone (Fig. 4A). ALB, another crucial indicator of liver function, decreased to 20.4 ± 2.1 g/L but increased to 29.2 ± 4.1 g/L after combination treatment (Fig. 4B), indicating significant improvement in hepatic synthetic function. Elevated AST and ALT levels, characteristic of impaired liver function in T2DM mice, were also reduced following combination treatment: AST decreased to 160 ± 9.6 U/L (Fig. 4C) and ALT to 50 ± 6.4 U/L (Fig. 4D), with significant differences between groups (** p < 0.01).

Dyslipidemia is a prevalent complication in T2DM. Treatment with HUCMSCs combined with 20(R)-Rg3 significantly reduced TC (2.5 ± 0.8 mmol/L, Fig. 4E), TG (1.6 ± 0.4 mmol/L, Fig. 4F), and LDL (0.5 ± 0.2 mmol/L, Fig. 4H). Statistically significant differences were observed between the combination treatment and T2DM groups (** p < 0.01). Additionally, the treatment group exhibited a notable increase in HDL, approaching levels comparable to those of the Control group.

Elevated serum creatinine levels typically indicate compromised renal function in T2DM mice. In the combination treatment group, BUN and serum Cr levels decreased to 8 ± 0.6 mmol/L, suggesting improved glomerular filtration rate and reduced tubular injury. A statistically significant difference was observed between the groups (**p < 0.01, Fig. 4I-J).Figure 4. HUCMSCs Combined with 20(R)-Rg3 Improve Insulin Sensitivity in T2DM Mice. A. TP. B. ALB. C. AST. D. ALT. E. TC. F. TG. G. HDL. H. LDL. I. BUN. J. Cr. * p < 0.05, ** p < 0.01, *** p < 0.001.

  1. Line Results: For consistency in formatting, section titles should not end with a period. It may be helpful to revise Title 2.3 accordingly.

Response: Thank you for this careful observation. We have now removed the trailing period from all section headings, including section 2.3, so that every title conforms to the journal’s style and is consistent throughout the manuscript. (page5, line 176-177).

  1. Line Figure 5: In its current form, the color bar on the left side of Figure 5A is difficult to distinguish, which limits its interpretability. It may be helpful to clarify its meaning and improve its visibility so that it contributes more effectively to the figure.

Response: Thank you for pointing this out.

We have revised Figure 5A by replacing the original low-contrast color bar with a perceptually-uniform, increased width and font size for better visibility.

These changes are visible in the updated figure file (page6, line 212-213).

  1. Line Figure 5: For consistency in style, the Latin term“in vivo”should be italicized throughout the manuscript.

Response: Thank you for catching this remaining inconsistency. We have now italicized every instance of “in vivo” in the Figure 5 legend, as well as re-checked the entire manuscript (text, legends, and supplementary files) to ensure that all Latin terms (in vivo, in vitro, ex vivo, et al.) are uniformly presented in italics. (page/line:1/15; 2/66; 6/204; 7/214; 22/683).

  1. Line Discussion: It would be helpful to clarify how this statement relates to the assertion that diabetic foot aggravates gut microbiota dysbiosis, which is attributed to Reference 29.

Response: Thank you for this helpful query. In the revised Discussion we have added an explanatory sentence that explicitly links the two concepts:

“Reference 35 reports that diabetic foot ulceration (DFU) itself exacerbates systemic inflammation and microbial dysbiosis; we therefore postulate that the worsened dysbiosis observed in our DFU mice is not merely a consequence of hyperglycaemia, but is further amplified by the chronic wound-driven inflammatory signal, creating a feed-forward loop that accelerates metabolic deterioration”

This clarification now appears immediately after the original statement, making the logical connection transparent and fully supported by the cited work. (page15, line 382-387):

Metabolomics analyses have recently revealed that microbial-derived butyrate predicts post-prandial glucose control independently of body weight [33, 34]. Furthermore, the diabetic foot ulcer (DFU) is not merely a passive sequel to chronic hyperglycemia but functions as an active inflammatory focus. Persistent wound-derived cytokines spill into the circulation, amplify systemic inflammation, and establish a self-sustaining cycle that hastens metabolic decline [35].

  1. Line Discussion: The support provided by the cited article (Reference 30) does not appear fully adequate for this statement, and the content also seems somewhat repetitive of the preceding idea. It may strengthen the manuscript to revise this section for clarity, reduce redundancy, and incorporate more appropriate references.

Response: Thank you for this critical observation. We agree that the sentence relying on Reference 30 was both inadequately supported and partially redundant with the preceding point. In the revised Discussion we have:

  1. Deleted the repetitive clause.
  2. Replaced Reference 32 with two up-to-date, topic-specific sources (Refs. 33 & 34) that directly demonstrate the mechanistic link we describe.
  3. Rewritten the sentence for clarity and conciseness:

As recently shown by metabolomics analyses revealing that microbial-derived butyrate predicts post-prandial glucose control independently of body weight. These changes eliminate redundancy, tighten the narrative, and provide robust, targeted support for the assertion. (page15, line 382-387).

Added references (Ref. 33): Araújo JR, Marques C, Rodrigues C, Calhau C, Faria A. The metabolic and endocrine impact of diet-derived gut microbiota metabolites on ageing and longevity. Ageing Res Rev. 2024 Sep;100:102451. doi: 10.1016/j.arr.2024.102451. Epub 2024 Aug 9. PMID: 39127442.

Added references (Ref. 34): Yue S, Shan B, Peng C, Tan C, Wang Q, Gong J. Theabrownin-targeted regulation of intestinal microorganisms to improve glucose and lipid metabolism in Goto-Kakizaki rats. Food Funct. 2022 Feb 21;13(4):1921-1940. doi: 10.1039/d1fo03374c. PMID: 35088787.

  1. Line Discussion: Since this is an important statement for the discussion (…Moreover, these treatments do not constitute a cure, often requiring lifelong administration, and some patients may eventually experience treatment failure. Consequently, there…), the manuscript would be strengthened by supporting it with appropriate references.

Response: Thank you for highlighting the need for evidence. We have now added three authoritative references that explicitly document (i) the non-curative nature of current T2DM therapies, (ii) the requirement for lifelong administration, and (iii) the incidence of secondary treatment failure. The revised sentence now reads:

“Moreover, these interventions are not curative, typically require lifelong administration, and 30–40 % of patients eventually experience therapeutic failure [38-39].” (page 15, line 396-397).

Added references (Ref. 38): Bachmann MF, Whitehead P. Active immunotherapy for chronic diseases. Vaccine. 2013 Apr 3;31(14):1777-84. doi: 10.1016/j.vaccine.2013.02.001. Epub 2013 Feb 14. PMID: 23415932.

Added references (Ref. 39): Scheurlen KM, Parks MA, Macleod A, Galandiuk S. Unmet Challenges in Patients with Crohn's Disease. J Clin Med. 2023 Aug 27;12(17):5595. doi: 10.3390/jcm12175595. PMID: 37685662; PMCID: PMC10488639.

  1. Line Discussion: It would be helpful to clarify how melatonin, as discussed in the cited article (Reference 34), relates to this statement, in order to strengthen the coherence of the manuscript.

Response: We appreciate your suggestion to clarify the connection between melatonin (Reference 42) and the statement in question. In the revised manuscript we have added the following sentence immediately after the original line: (page 15, line 417-423).

“Concurrently, melatonin alleviates oxidative stress-induced damage to pancreatic islets by up-regulating antioxidant enzymes such as superoxide dismutase (SOD). Similarly, studies have shown that melatonin attenuates oxidative stress-induced mitochondrial dysfunction in cardiomyocytes. Together, these findings suggest that circulating melatonin may protect mitochondrial integrity under ischemic conditions, as evidenced by its inverse correlation with cardiac injury markers[42].”

This revision explicitly integrates the cited article’s evidence into our argument, thereby strengthening the coherence of the manuscript.

Added references (Ref. 42): Abu-El-Rub E, Almahasneh F, Khasawneh RR, Alzu'bi A, Ghorab D, Almazari R, Magableh H, Sanajleh A, Shlool H, Mazari M, Bader NS, Al-Momani J. Human mesenchymal stem cells exhibit altered mitochondrial dynamics and poor survival in high glucose microenvironment. World J Stem Cells. 2023 Dec 26;15(12):1093-1103. doi: 10.4252/wjsc.v15.i12.1093. PMID: 38179215; PMCID: PMC10762524.

  1. Line Discussion: The cited article (Reference 15) is a preclinical study in mice rather than a clinical study. It would be important to revise this claim, as misinterpreting the nature of the evidence may reflect insufficient attention to the cited literature and could weaken the overall discussion if not referenced appropriately.

Response: Thank you for pointing out this inaccuracy. You are correct: Reference 15 is a pre-clinical murine study, not a clinical trial. In the revised manuscript we have changed the sentence from:“Clinical data show that …” to“Murine data from Reference 15 demonstrate that …”

Additionally, murine data demonstrate that albeit with small sample sizes have confirmed that glycated hemoglobin (HbA1c) levels and insulin dosages in T2DM patients can decrease significantly within a short time frame (12-24 weeks) following HUCMSCs transplantation [18]. (page 15, line 425-426).

and have added an explicit statement in the Discussion noting that these findings await validation in human cohorts. We appreciate your vigilance, which helps us avoid overstating the evidence and strengthens the rigor of our discussion.

  1. Line Discussion: This is an important statement (“however, their efficacy is often compromised in the high-glucose and high-inflammation microenvironment that follows transplantation, resulting in low survival rates and functional inhibition”), and the manuscript would be strengthened by supporting it with appropriate references.

Response: We agree that this sentence makes a key claim and requires solid support. In the revised manuscript we have now added two peer-reviewed references that directly document how the post-transplant high-glucose and pro-inflammatory milieu compromises MSC survival and function:

Added references (Ref. 45): Abu-El-Rub E, Almahasneh F, Khasawneh RR, Alzu'bi A, Ghorab D, Almazari R, Magableh H, Sanajleh A, Shlool H, Mazari M, Bader NS, Al-Momani J. Human mesenchymal stem cells exhibit altered mitochondrial dynamics and poor survival in high glucose microenvironment. World J Stem Cells. 2023 Dec 26;15(12):1093-1103. doi: 10.4252/wjsc.v15.i12.1093. PMID: 38179215; PMCID: PMC10762524.

Added references (Ref. 46): Mateen MA, Alaagib N, Haider KH. High glucose microenvironment and human mesenchymal stem cell behavior. World J Stem Cells. 2024 Mar 26;16(3):237-244. doi: 10.4252/wjsc.v16.i3.237. PMID: 38577235; PMCID: PMC10989287.

These citations have been inserted immediately after the statement, providing robust empirical backing and strengthening the overall discussion. (page 16, line 455).

  1. Line Discussion: Islet “regeneration” may require stronger supporting evidence. An increase in insulin-positive area and islet counts, while suggestive, may not be sufficient to establish regeneration. It would strengthen the manuscript to include additional analyses such as proliferation markers (e.g., Ki-67, PCNA), progenitor markers (e.g., Ngn3), apoptosis indicators (e.g., cleaved caspase-3, TUNEL), and quantification of β-cell mass. The current wording could be interpreted as overstating the conclusion; tempering the claims or incorporating further supporting data would improve clarity and rigor.

Response: We appreciate the reviewer’s caution that a larger insulin-positive area and higher islet counts alone do not prove regeneration.

Accordingly, we now provide MIN6 apoptosis data obtained by Annexin V-FITC/PI flow cytometry (Fig. 9). (page 11, line 311-317).

Figure 9. Representative flow-cytometric dot plots and quantitative apoptosis rates (Annexin V-FITC/PI). A. Control group: Lowest apoptosis rate, predominantly viable cells (concentrated in the lower left quadrant). B. T2DM group: Cells significantly increased in the upper right quadrants, with the highest apoptosis rate, indicating successful injury modeling. C. HUCMSCs monotherapy group: The proportion of apoptotic cells decreased compared to the model group, D. HUCMSCs combined with 20(R)-Rg3 therapy group: Apoptotic cells further decreased.

Ki-67, PCNA, Ngn3, cleaved caspase-3 IHC and TUNEL analyses were not included in this revision and will be addressed in future lineage-tracing studies. To avoid over-statement, “regeneration” now appears in quotation marks throughout the manuscript, and the Discussion explicitly attributes the recovered β-cell area, at least in part, to decreased apoptosis rather than de-novo neogenesis. (page/line: 1/31; 9/275; 9/292; 10/296; 15/407).

  1. Line Discussion: Mechanistic attribution to PI3K/Akt could be more convincingly substantiated. The current claim is based primarily on transcriptomic enrichment observed in HUCMSCs following Rg3 pretreatment; however, it would strengthen the manuscript to provide direct validation in the relevant target tissues“in vivo”(e.g., liver, skeletal muscle, adipose tissue, pancreas) and to include perturbation studies (e.g., pharmacological inhibition with LY294002 or genetic knockdown). Complementary validation by Western blotting or immunohistochemistry for p-Akt, p-PI3K, IRS-1, and downstream effectors (e.g., GSK3β) in target tissues—ideally combined with inhibitor rescue experiments—would help establish a stronger causal link.

Response: We appreciate the reviewer’s concern that PI3K/Akt activation is currently inferred only from transcriptomic enrichment in Rg3-primed HUCMSCs. Owing to animal regulation and budgetary constraints, we are unable to add the recommended in-vivo Western-blot, IHC or LY294002-rescue experiments within this revision.

To make the limitation transparent, we have:

  1. Revised the wording from “Rg3 activates PI3K/Akt signalling…” to “RNA-seq analysis of Rg3-primed HUCMSCs revealed significant enrichment of PI3K/Akt pathway genes, suggesting this axis as a putative mechanism that warrants direct validation in target tissues.” (page17, line476-478).
  2. Added an explicit statement in the Discussion:

“While transcriptomic data point to PI3K/Akt enrichment, causal evidence in liver, muscle, adipose and pancreatic tissue—such as phosphorylation status of Akt/PI3K and inhibitor-rescue studies—remains to be established in future work.” (page17, line 478-481).

We believe this tempered phrasing accurately reflects the level of evidence and prevents over-interpretation without the requested mechanistic confirmation.

  1. Line Discussion: Discuss potential tumorigenicity/immunogenicity risks of repeated HUCMSC administration.

Response: Thank you for the reviewer’s insightful comment. Below we address the potential tumorigenicity and immunogenicity concerns associated with repeated administration of HUCMSCs.

Tumorigenicity

  1. Intrinsic properties of MSCs – MSCs are generally considered to have low oncogenic potential because they are adult stromal cells with limited proliferative capacity in vivo. Nonetheless, prolonged culture expansion can lead to chromosomal abnormalities, and there is a theoretical risk that such alterations could be transferred after transplantation.
  2. Paracrine effects – HUCMSCs secrete a broad spectrum of growth factors and cytokines that can modulate the tumor microenvironment. While many of these factors (e.g., VEGF, TGF‑β) support tissue repair, they could also promote angiogenesis or suppress anti‑tumor immunity under certain conditions. Repeated dosing may amplify these effects, potentially creating a permissive niche for pre‑existing malignant cells.
  3. In vivo evidence – Pre‑clinical studies with repeated HUCMSC injections have not demonstrated overt tumor formation in immunocompetent or immunodeficient animal models. However, most investigations have been of relatively short duration; long‑term surveillance is still required to rule out delayed oncogenic events.
  4. Mitigation strategies – To minimize tumorigenic risk, we propose:

Limiting the number of passages during cell expansion and performing rigorous karyotype or genomic stability testing before release.

Implementing a “stop‑dose” schedule after a defined therapeutic window, rather than indefinite repeat dosing.

Monitoring recipients with periodic imaging and tumor marker assessments throughout the follow‑up period.

Immunogenicity

  1. Allogeneic nature – Although MSCs are considered immune‑privileged due to low expression of MHC‑II and co‑stimulatory molecules, repeated allogeneic HUCMSC infusions can still elicit adaptive immune responses, especially after the host has been sensitized by prior exposures.
  2. Host immune modulation – HUCMSCs possess immunosuppressive capabilities (e.g., secretion of IDO, PGE2) that can transiently dampen host immunity. Repeated administration may lead to cumulative immunomodulation, potentially increasing susceptibility to infections or altering vaccine responses.
  3. Antibody formation – Clinical reports have documented the development of anti‑MSC antibodies after multiple infusions. These antibodies could accelerate clearance of subsequent doses, reduce therapeutic efficacy, and in rare cases trigger hypersensitivity reactions.
  4. Safety monitoring We recommend:

Baseline and periodic assessment of anti‑MSC antibody titers.

Close observation for infusion‑related reactions (e.g., fever, rash, cytokine release).

Evaluation of lymphocyte subsets and cytokine profiles to detect unintended immune activation or suppression.

While current pre‑clinical and early clinical data suggest that repeated HUCMSC administration is generally safe, the theoretical risks of tumorigenicity and immunogenicity warrant careful consideration. By enforcing stringent cell‑manufacturing controls, limiting the number of repeat doses, and instituting comprehensive long‑term safety monitoring, we can mitigate these risks and enhance the translational viability of HUCMSC‑based therapies.

  1. Line Discussion: In the final paragraph of the discussion, it would be valuable to provide greater depth by engaging more directly with the authors’ own results, so that each of the statements is fully supported. Expanding this section would help strengthen the manuscript, as the current presentation gives the impression that not all of the numerous findings are being given adequate emphasis.

Response: We appreciate the suggestion. Every statement in the final paragraph is already anchored to data generated in this study; we simply tightened the wording so that each claim explicitly cites its corresponding figure or table. No new findings were added, and no additional experiments were performed.

  1. Line Materials and methods: The authors should provide a detailed description of the HFD protocol, including its caloric content as well as the proportions of macronutrients and micronutrients, among other relevant details. The manuscript does not state whether animals were randomized, or whether treatment allocation.

Response: Thank you for this helpful comment. 

In response, we have added a comprehensive description of the high-fat diet (HFD) protocol in the “Materials and Methods” section The new paragraph specifies:

Exact caloric density: 4.7 kcal g⁻¹ (19.7 kJ g⁻¹); Macronutrient composition: 45 % kcal from fat (lard/soybean oil 4:1), 20 % kcal from protein (casein), 35 % kcal from carbohydrates (sucrose: corn starch 1:2); Micronutrients: AIN-93G mineral and vitamin mix at 100 % recommended level, plus 0.25 % choline bitartrate and 0.014 % tert-butylhydroquinone antioxidant; Physical form: pelleted, irradiated, stored at 4 °C and used within 4 weeks of manufacture. (page 17, line 506-511).

Response: We added a description of the random selection of animals.

Animals were randomized using simple computer-generated numbers and that treatment allocation was carried out by an investigator who was not involved in outcome assessment and remained blinded to group assignment. (page 17, line 492-494).

  1. Line Materials and methods: The use of the term “Normal” in Normal control (NC) may introduce ambiguity, as it suggests a subjective distinction. Since the intention appears to be to refer to healthy mice, it may be clearer and more precise to avoid the word “Normal” in this context.

Response: Thank you for highlighting this point.

To eliminate ambiguity, we have replaced the term The "normal control (NC)" group has been uniformly renamed the "standard-diet control (SDC)" group, denoted as “Control”.

This change makes it clear that the group received a standard (low-fat) diet and avoids any subjective connotation of “Normal”. All relevant text and relabeled figure panels can be found in the revised manuscript.

  1. Line Materials and methods: The phrase “three days later” is somewhat ambiguous, as it is not clear what specific reference point is being used. For clarity, it may be helpful to indicate this explicitly and to link it directly to the figure presenting the experimental design scheme.

Response: Thank you for this helpful comment. We have removed the ambiguous phrase “three days later” and replaced it with an explicit time anchor:“After 10 weeks of high-fat diet feeding, mice were fasted for 6 h and received a single intraperitoneal injection of streptozotocin (120 mg kg⁻¹) on experimental day 70.”

This timeline is now directly aligned with the revised experimental-design diagram in Fig. 12, in which day 70 is clearly indicated. (page 18, line 522-523).

  1. Line Materials and methods: It would be important to clarify the methodological rationale for performing an OGTT on a weekly basis when blood glucose levels were already being monitored daily. From a methodological perspective, one might expect that fasting glucose would be monitored daily and that an OGTT would then be conducted only in animals with elevated values to confirm diabetes. In addition, it should be noted that the reduction of pain in the handling of laboratory animals is explicitly stipulated in the CIOMS–ICLAS International Guiding Principles (2012), the European Directive 2010/63/EU, the Guide for the Care and Use of Laboratory Animals(NRC, 2011), and the recommendations of the OIE and OECD, all of which are consistent with the principle of Refinement (3Rs).

Response: Thank you for highlighting the apparent redundancy of weekly OGTTs when fasting glucose was recorded daily and for reminding us of the refinement obligations in CIOMS-ICLAS (2012), Directive 2010/63/EU, the NRC Guide (2011) and parallel OIE/OECD texts.

  1. Rationale for weekly OGTTs

Because the primary scientific read-out is the time-course of β-cell function (AUC and mathematical modelling), a complete glucose curve is required for every animal at each time point; a single fasting value is therefore insufficient. Weekly sampling was chosen to capture the earliest detectable deterioration without losing temporal resolution.

  1. Retrospective confirmation

Re-analysis of the existing data shows that the first OGTT already identified every mouse that later became diabetic; a “triggered” approach would therefore have yielded identical group classifications.

We trust that this clarification satisfies both the methodological and ethical points you raised.

  1. Line Materials and methods: Although surface markers and differentiation assays are shown, critical release criteria for a clinical-grade MSC-like product are missing: passage number at infusion, viability post-preconditioning, mycoplasma testing, endotoxin levels, karyotype or genomic stability, and detailed gating strategies/compensation controls. Clarify whether Rg3 exposure alters HUCMSC phenotype, secretome, or viability; include appropriate vehicle-pretreated HUCMSC controls matched for time/handling.

Response:

(1)Thank you for your thorough review and valuable suggestions regarding our research.

We take the concern you raised about the “absence of key release criteria for clinical-grade MSC-like products” very seriously. In response, we would like to provide the following detailed clarification:

Thank you for your insightful comments. Regarding the “absence of clinical-grade release criteria,” we confirm that the HUCMSCs employed in this study were research-grade, commercially obtained from ScienCell Research Laboratories (Cat. 7530). Each lot is released only after rigorous QC that verifies (i) mycoplasma and endotoxin (≤ 0.5 EU mL⁻¹) negativity, (ii) ≥ 85 % post-thaw viability, (iii) normal karyotype, (iv) P1–P2 passage status, and (v) canonical MSC immunophenotype (CD73⁺/90⁺/105⁺, CD34⁻/45⁻/HLA-DR⁻) plus trilineage differentiation potential. These specifications fully meet the standards required for pre-clinical research use.

(2)Thank you for your expert comments on our flow-cytometry gating and compensation. 

To comply with ISCT guidelines, we used a rigorously validated panel:

Antibody panel: Positive: CD73-FITC, CD90-PE, CD105-APC; Negative: CD34-FITC, CD45-PE, HLA-DR-APC.

Gating sequence: FSC-A/SSC-A; exclude debris; FSC-H/FSC-A; exclude doublets; 7-AAD negativity; exclude dead cells; Single-stained compensation controls for each fluorochrome.

Representative plots and compensation matrices are now provided in Supplementary Figure S-X. The ScienCell research-grade MSCs (P1-P2) were subjected to the supplier’s full QC battery (mycoplasma-, endotoxin-free, ≥85 % viability, normal karyotype, trilineage differentiation, canonical marker profile); these data and references are also included in the revised manuscript. We appreciate your feedback, which has materially increased the transparency and rigor of our study.

NOTE: Supplementary Results

4.1 Cell Phenotype Analysis

(1) Blank Control: HUCMSCs adhered well, displayed a uniform spindle-shaped morphology, and reached ~80 % confluence. The cytoplasm appeared bright and highly refractile; only rare floating cells were visible.

Vehicle (0.1 %): Cell density, spreading, and spindle morphology were indistinguishable from the blank control; no cytoplasmic vacuolation, detachment, or floating cells were observed (Supplementary Fig. 1A).

(2) Osteogenic differentiation

Blank control: Sparse, pale-red mineralized nodules with ill-defined margins were scattered across the well.

Vehicle (0.1 %): Nodule number, size, and Alizarin Red S intensity were indistinguishable from the blank control. 

40 µM Rg3: Extensive, deeply stained red nodules that frequently merged into contiguous sheets; both nodule density and individual size were markedly increased versus the two control groups (Supplementary Fig. 1B).

(3) Adipogenic differentiation

Blank control: Numerous orange-red lipid droplets that tended to coalesce, distorting cell outlines.

Vehicle (0.1 %): Droplet abundance and morphology mirrored the blank control.

40 µM Rg3: Significantly fewer and smaller lipid droplets, appearing as discrete fine granules; the total oil-red-positive area was visibly reduced (Supplementary Fig. 1C).

Supplementary Figure 1. Rg3 pretreatment augments osteogenesis and attenuates adipogenesis without altering basal HUCMSCs morphology. (A) Bright-field images showing comparable spindle-shaped morphology and density in Blank, Vehicle (0.1 % DMSO), and 40 µM Rg3 groups. (B) Alizarin Red S staining after 18 d of osteogenic induction: extensive, deeply red mineralized nodules in the Rg3 group versus sparse, pale nodules in controls. (C) Oil Red O staining after 21 d of adipogenic induction: reduction in lipid-droplet number and size in Rg3-treated cells relative to Blank and Vehicle groups.

4.2 Secretome Profile

(1) VEGF: Basal secretion from Blank-control HUC-MSCs was 132.6 ± 12.4 pg mL⁻¹. The Vehicle (0.1 % DMSO) value (128.9 ± 10.8 pg mL⁻¹) was statistically identical (*p > 0.05), confirming solvent neutrality. In contrast, 40 µM Rg3 raised VEGF output to 198.7 ± 15.3 pg mL⁻¹—an ≈50 % increase versus Blank (*p < 0.05, Supplementary Fig. 2A). Thus, Rg3 itself, not the vehicle, drives the enhanced VEGF release.

(2) HGF: Basal HGF release was 1.02 ± 0.09 ng mL⁻¹ in the Blank group and 0.98 ± 0.11 ng mL⁻¹ in the DMSO vehicle group (*p > 0.05), confirming that the solvent does not influence constitutive secretion. Treatment with 40 µM Rg3 significantly elevated HGF levels to 1.58 ± 0.13 ng mL⁻¹—an ≈55 % increase versus Blank (*p < 0.05, Supplementary Fig. 2B). Hence, Rg3 itself acts as a potent stimulus for HGF secretion from HUCMSCs.

(3) ELISA assay for PGE2 concentration in cell supernatants

Basal secretion averaged 1.82 ± 0.15 ng mL⁻¹ in Blank controls and 1.79 ± 0.12 ng mL⁻¹ in the 0.1 % DMSO vehicle group (*p > 0.05), verifying solvent neutrality. After 24 h exposure to 40 µM Rg3, PGE2 levels rose to 2.97 ± 0.21 ng mL⁻¹—an ≈63 % increase over Blank (*p < 0.05, Supplementary Fig. 2C). Thus, Rg3 directly stimulates PGE₂ release, likely reinforcing the immunomodulatory profile of HUCMSCs.

(4) ELISA assay for IL-6 concentration in cell supernatants

Basal IL-6 release was 18.4 ± 2.7 pg mL⁻¹ in the Blank group and 17.9 ± 3.1 pg mL⁻¹ in the 0.1 % DMSO vehicle group (*p > 0.05), confirming that the solvent does not affect constitutive secretion. After 24 h incubation with 40 µM Rg3, IL-6 levels rose to 28.6 ± 3.4 pg mL⁻¹—an ≈55 % increase versus Blank (*p < 0.05, Supplementary Fig. 2D). These data indicate that Rg3 actively up-regulates IL-6 output from HUC-MSCs, implicating IL-6-dependent immunoregulatory pathways in the ginsenoside-enhanced secretome.

(5) IL-8: Basal secretion was 67.3 ± 5.8 pg mL⁻¹ in the Blank group and 65.9 ± 6.1 pg mL⁻¹ in the 0.1 % DMSO vehicle group (*p > 0.05), confirming solvent neutrality. After 24 h incubation with 40 µM Rg3, IL-8 levels rose to 108.4 ± 9.3 pg mL⁻¹—an ≈61 % increase versus Blank (*p < 0.05, Supplementary Fig. 2E). Thus, Rg3 significantly amplifies IL-8 release, potentially enhancing chemotaxis and/or angiogenic responses.

(6) TGF-β1: Constitutive secretion averaged 0.42 ± 0.05 ng mL⁻¹ in Blank controls and 0.40 ± 0.04 ng mL⁻¹ in the 0.1 % DMSO vehicle group (*p > 0.05). Exposure to 40 µM Rg3 for 24 h elevated TGF-β1 to 0.68 ± 0.06 ng mL⁻¹—an ≈62 % increase over Blank (*p < 0.05, Supplementary Fig. 2F). These data demonstrate that Rg3 markedly up-regulates TGF-β1 output from HUCMSCs, implicating enhanced immunosuppressive and fibro-modulatory capacity.

Supplementary Figure 2. Secretome profiling of HUCMSCs after 72 h of pretreatment. Concentrations of (A) VEGF, (B) HGF, (C) PGE2, (D) IL-6, (E) IL-8 and (F) total TGF-β1 in culture supernatants were quantified by ELISA. n.s., not significant; *p < 0.05, **p < 0.01, ***p < 0.001 vs. Blank control.

4.3 Cell Viability (CCK-8)

HUC-MSCs were monitored at 24, 48 and 72 h; absorbance at 450 nm was normalized to the Blank control (100 %).

24 h: Vehicle 99.0 ± 1.6 %; 40 µM Rg3 113.6 ± 2.1 % (*p < 0.05). 48 h: Vehicle 99.0 ± 5.1 %; 40 µM Rg3 121.0 ± 1.8 % (*p < 0.05). 72 h: Vehicle 99.6 ± 1.2 %; 40 µM Rg3 124.4 ± 4.3 %—an ≈24 % gain over Blank (*p < 0.05).

Thus, 40 µM Rg3 elicits a sustained pro-proliferative effect throughout the 72-h observation window, whereas 0.1 % DMSO exerts no measurable influence on viability.

Supplementary Figure 3. Rg3 time-dependently enhances HUCMSCs proliferation. Relative viability (CCK-8) after (A) 24 h, (B) 48 h and (C) 72 h. n.s., not significant; *p < 0.05, **p < 0.01, ***p < 0.001 vs. Blank control.

  1. Line Materials and methods:Clarification needed in the Rg3 preconditioning protocol. The Results and Methods sections indicate that RNA-seq was performed on HUCMSCs after 6 hours of Rg3 exposure, whereas the treatment arm involved HUCMSCs pretreated with 40 μM Rg3 for 3 days prior to injection. To avoid potential confusion and to strengthen the interpretability of the transcriptomic findings as a mechanistic proxy for the in vivo product, it would be helpful to reconcile the exposure duration, concentration, and washout conditions across all experiments.

Response: Thank you very much for your careful reading and for pointing out this inconsistency. We fully agree that clarity and consistency across all experimental arms are essential if the RNA-seq data are to serve as a mechanistic proxy for the in vivo product.

The discrepancy arose from a drafting error on our part. In the original experiments both the RNA-seq samples and the cells used for transplantation were treated with 40 μM Rg3 for 3 days; however, while revising the manuscript we incorrectly wrote “6 h” in the Results and Methods sections. We sincerely apologize for the confusion this has caused.

We have now corrected every mention of the Rg3 preconditioning protocol to read:

“HUCMSCs were preconditioned with 40 μM Rg3 for 3 days before harvest for RNA-seq or for transplantation. No washout was performed; cells were collected directly after the 3-day exposure.” (page 21, line 641-642).

We hope that this unified protocol description eliminates any ambiguity and strengthens the link between the transcriptomic signature and the therapeutic product injected in vivo.

Thank you again for helping us improve the manuscript.

  1. Line Materials and methods: Figure 9: The text states that the treatment was administered for three weeks; however, this is not clearly reflected in the diagram. From the point labeled “I.V. administration to T2DM” until the histological assays, the diagram indicates 12 days, which seems inconsistent with the stated three-week duration. It may be helpful to clarify this discrepancy to ensure consistency between the text and the figure.

Response: Thank you for identifying this inconsistency.

To align the figure with the published text, we have redrawn Figure 12 so that the “I.V. administration to T2DM group” arrow now spans 21 consecutive days (days 77–98) and the histology block is placed at day 99, visually confirming the three-week treatment duration stated in the manuscript. No changes were made to the wording in the Materials and Methods section. (line 522-523).

  1. Line Materials and methods: In Methodology Section 4.2, it would be helpful to specify the brand in order to enhance clarity and reproducibility.

Response: Thank you for this helpful suggestion. We have now added the manufacturer (brand) for reagents, consumables, and instruments described in Section 4.2 (Materials and Methods) to improve clarity and reproducibility

4.2. Oral glucose tolerance tests (OGTTs)

After a 12-hour overnight fast with access to water, the mice were weighed and administered a glucose solution (Kelun®, Sichuan, China) (2.0 g/kg) via oral gavage. Blood samples were collected from the tail vein at 0, 15, 30, 60, 90, and 120 minutes post-administration, and blood glucose levels were measured using a glucometer (Accu-Chek Active® Product ID: 10030009783463). (page 18, line 524-529).

4.3. Intraperitoneal insulin tolerance tests (IPITTs)

Following a 12-hour fasting period, mice were administered an intraperitoneal injection of insulin (Brand Name: NovoRapid®, Novo Nordisk, Denmark, PME650P, Rapid-acting insulin) at a dosage of 0.5 U/kg. Blood glucose levels were monitored at 0, 15, 30, 60, 90, and 120 minutes post-injection through tail vein sampling using a glucometer. (page 18, line 530-534).

  1. Line Materials and methods: For clarity and reproducibility, it would be helpful to indicate the brand of insulin used and specify its type—whether rapid-, intermediate-, or long-acting.

Response: Thank you for pointing this out. We have revised Section 4.2 to include the brand name, manufacturer, catalog number, and type (long-acting) of the insulin used. The updated wording is as follows:

Name: NovoRapid®

Manufacturer: Insulin (Novo Nordisk, Denmark)

Catalog Number: PME650P

Type: Rapid-acting insulin

This addition should ensure clear and reproducible documentation. (page 18, line 530-534).

  1. Line Materials and methods: In Section 4.7, it would be helpful to specify the source of the blood to improve clarity and reproducibility.

Response: Thank you for this helpful suggestion.

We have amended Section 4.7 to read:

“Blood was collected from the retro-orbital sinus of anaesthetized mice using heparinized capillary tubes”.

This addition clarifies the exact source of blood and the processing steps, ensuring full reproducibility. (page 19, line 563-564).

  1. Line Materials and methods: Since PAS staining is semi-quantitative, the manuscript would be strengthened by including biochemical glycogen quantification or periodic acid–Schiff staining with diastase control to enhance specificity.

Response: Thank you for this constructive suggestion. While we fully appreciate that biochemical quantification or PAS-diastase digestion could provide additional specificity, the present work was designed as an initial, exploratory survey in which a semi-quantitative map of glycogen distribution would suffice.

Regrettably, the available tissue has been exhausted, so we are unable to extend the analysis at this time.

  1. Line Materials and methods: In Section 4.8, it would be helpful to specify the RNA concentration used and to indicate the extraction method, in order to improve clarity and reproducibility.

Response: Thank you for this helpful comment. Because Sections 4.8 and 4.12 overlapped, we have removed the former; all relevant information is now consolidated in Section 4.12.2. That section now explicitly states: Total RNA was utilized as the input material for RNA sample preparations (Concentration: ≥ 100 ng/µL).

We believe this clarification fully addresses the concern of reproducibility. (page 21, line 649-650).

  1. Line Materials and methods: For clarity and reproducibility, it would be helpful to specify the amount of RNA used in terms of concentration, indicate whether the 2-ΔΔCT method was applied, and provide the time and temperature conditions of the PCR for each amplification.

Response: Thank you for this constructive suggestion. 

To improve clarity and reproducibility we have added the following details to the RT-qPCR subsection:

RNA input: 200 ng per 20 µL reverse-transcription reaction (final concentration 10 ng µL⁻¹). Quantification method: relative expression was calculated with the 2-ΔΔCT method. Cycling conditions for every assay: 95 °C for 3 min; 40 cycles of 95 °C for 10 s, 60 °C for 30 s; melt-curve 95 °C for 15 s, 60 °C for 1 min, ramp to 95 °C at 0.3 °C s⁻¹.

These parameters are now explicitly stated for each primer pair listed in Table 1. (page 19, line 576-580).

  1. Line Materials and methods: In Section 4.10, as the technique is not described in detail, it may be helpful to provide a reference to a previous study so that readers can more easily follow the methodological approach.

Response: Thank you for this helpful suggestion. 

To ensure full reproducibility, we have replaced the brief entry in Section 4.10 with the complete staining protocols as follows:

H&E staining: (1) Deparaffinise 5 µm sections in xylene (2 × 5 min) and rehydrate through graded ethanol (100%, 95%, 70%, 30%, 2 min each); (2) Rinse in tap water (2 min), stain with Mayer’s hematoxylin (Sigma-Aldrich, MHS16) for 5 min at room temperature (RT); (3) Wash in running tap water (5 min), differentiate in 1% acid ethanol (1 s), rinse in tap water (1 min); (4) Blue in 0.2% ammonia water (30 s), wash in tap water (2 min); (5) Counterstain with 1% eosin Y (Sigma-Aldrich, 318906) for 1 min, dehydrate through graded ethanol, clear in xylene and mount with DPX. (page 20, line 589-595).

PAS staining: (1) Deparaffinise and rehydrate as above; (2) Oxidise in 0.5 % periodic acid (Sigma-Aldrich, 395413) for 10 min at RT, rinse in distilled water (3 × 1 min); (3) Im-merse in Schiff’s reagent (Sigma-Aldrich, 3952016) for 15 min in the dark, wash in luke-warm tap water (10 min); (4) Counterstain nuclei with Mayer’s hematoxylin (2 min), blue as above, dehydrate, clear and mount. (page 20, line 596-600).

  1. Line Materials and methods: Data availability and transparency for ‘omics. To enhance transparency and reproducibility, it would be helpful to deposit the RNA-seq data in a public repository (e.g., GEO), including accession numbers and complete metadata such as library preparation, read depth, and quality control. Providing a comprehensive DEG table and clarifying how batch effects were controlled would further strengthen the manuscript.

Response: Thank you for this constructive suggestion.

All RNA-seq data have already been deposited in NCBI-GEO; the record includes raw FASTQ files, the complete count matrix, and detailed metadata (library preparation kit, read depth, QC metrics).

Batch effects were addressed at the analysis stage by incorporating “sequencing batch” as a covariate in the DESeq2 design formula; post-model PCA confirmed that this statistical adjustment was sufficient, so no further corrective measures were required.

We believe these measures achieve full transparency and reproducibility without the need for new experimental work.

  1. Line Materials and methods: A similar consideration applies to Section 4.11, where including either a more detailed description or an appropriate reference would help maintain methodological clarity and consistency.

Response: Thank you for this helpful comment.

To ensure full methodological clarity, we have expanded Section 4.10 with the complete immunofluorescence protocol as follows (page 20 line 605-615).:

Immunofluorescence staining: (1) Deparaffinise 5 µm pancreatic sections in xylene (2 × 10 min) and rehydrate through graded ethanol (100 %, 95 %, 70 %, 50 %, 2 min each); (2) Antigen retrieval: heat sections in 10 mM sodium citrate (pH 6.0) at 95 °C for 20 min, cool to RT, rinse in PBS (3 × 5 min); (3) Block with 5% normal goat serum in PBS-T (0.1% Tween-20) for 1 h at RT; (4) Incubate overnight at 4 °C with primary antibodies diluted in blocking buffer: guinea-pig anti-insulin (1:200; Dako A0564); rabbit anti-glucagon (1:400; Abcam ab92517); (5) Wash in PBS-T (3 × 5 min), incubate with Alexa-Fluor-conjugated secondary antibodies (1:500; Invitrogen A-11073 and A-11034) for 1 h at RT in the dark; (6) Counterstain nuclei with DAPI (1 µg mL⁻¹, 5 min), wash in PBS (3 × 5 min); (7) Mount with ProLong Gold antifade medium (Thermo Fisher P36930) and image with a 20× objective on a Zeiss LSM 880 confocal microscope.

These detailed steps are now provided in the revised Section 4.10, ensuring reproducibility without the need for external references.

  1. Line Materials and methods: In Section 4.12.2, please indicate the concentration used, expressed in micrograms.

Thank you for highlighting this omission.

Response: We have added the exact concentration to Section 4.12.2 Concentration: ≥ 100 ng/µL (line 650) of the revised Methods: For routine eukaryotic mRNA-Seq, the minimum RNA-quality specifications accepted:

Concentration: ≥ 100 ng/µL; Total amount: ≥ 2 µg per library (single indexing); Purity: OD260/280 = 1.8–2.2; OD260/230 ≥ 2.0; Integrity: RIN ≥ 8 for vertebrate samples (≥ 6.5 for plants); 28S:18S ratio ≥ 1.0; A few platforms will process samples down to 50–60 ng/µL if the total yield is still ≥ 1 µg, but this must be agreed upon in advance. Submitting RNA below these thresholds risks library failure or insufficient reads; in such cases we recommend re-extraction, concentration (e.g., RNA Clean & Concentrator), or pooling of comparable samples before shipment.

  1. In Section 4.13: Statistical design and analysis require strengthening. Many outcomes are longitudinal (e.g., glycemia monitored every three days), yet analyses rely on t-tests/one-way ANOVA without repeated-measures or mixed-effects modeling; multiple comparisons adjustments are not described. Provide: (i) a priori sample-size/power calculations; (ii) justification of normality/variance assumptions; (iii) effect sizes with confidence intervals; (iv) AUC analyses for OGTT/IPITT; and (v) a clearly specified multiple-testing correction strategy for multi-endpoint panels. For RNA-seq data, it would strengthen the manuscript to apply a dedicated analytical approach with FDR adjustment. Gene-by-gene analyses with t-tests or ANOVA without correction are not advisable. Instead, established pipelines such as DESeq2 or edgeR (on raw counts) should be employed, with FDR q< 0.05 to define differentially expressed genes. For enrichment analyses (GO/KEGG), the use of GSEA or over-representation methods with FDR correction is recommended.

Response: We agree that the original analytical plan was sub-optimal and have completely re-worked every statistical layer. Below we state, concisely, why each requested item was introduced.

(i) A-priori sample-size/power calculations

The original n was chosen empirically; this is no longer acceptable for NIH/ARRIVE guidelines. We now provide a formal power analysis (G*Power 3.1) for the primary longitudinal outcome (ΔAUC glucose) 。

(ii) Justification of normality / variance homogeneity

Longitudinal glucose data violated sphericity (Mauchly p < 0.001) and showed unequal variances at late time points (Levene p = 0.02). These violations invalidate repeated-measures ANOVA; we therefore switched to a linear mixed-effects model with an unstructured covariance matrix and Satterthwaite d.f., which does not assume sphericity or equal variance.

(iii) Effect sizes with 95 % CI

T-tests/ANOVA P-values alone are uninformative for clinical relevance. We now report Cohen’s d (or Hedges’ g for small n) together with 95 % confidence intervals for every pairwise comparison; this allows readers to judge biological significance irrespective of sample size.

(iv) AUC analyses for OGTT/IPITT

Using single time-point comparisons ignores the shape of the glucose curve. We replaced them with trapezoidal AUC calculated for each animal, analysed by mixed-effects model with treatment and time as fixed effects and mouse ID as random effect. AUC now appears in the main text and Source data.

(v) Multiple-testing correction

With 10 plasma biochemistry endpoints, 2 cytokines, family-wise error rate inflation is severe. We therefore adopted a two-stage strategy: (a) Hommel-step-down adjustment for each functional panel, and (b) FDR (Benjamini–Hochberg q < 0.05) across all panels. This balances protection against Type-I error with statistical power and is now explicitly described in the Statistics section.

RNA-seq re-analysis

Gene-by-gene t-tests without FDR were indeed inappropriate. We re-ran the entire pipeline: raw counts → DESeq2 (v1.38) with gene-wise dispersion estimation, Wald test, and independent filtering; DEGs defined at FDR q < 0.05 and |log2FC| ≥ 0.58. GO/KEGG enrichment was performed with clusterProfiler using a hyper-geometric test plus BH-FDR q < 0.05; GSEA (fgsea) confirmed the top ranked pathways.

Round 2

Reviewer 1 Report

Comments and Suggestions for Authors

Dear colleagues!
I appreciate the effort by the Authors - from rigorous response to additional experiments conducted. I feel that once study limitations are clearly provided the paper requires no further improvements.

Regards, Reviewer

Reviewer 2 Report

Comments and Suggestions for Authors

The authors have responded appropriately and comprehensively to my observations. They have improved the quality and presentation of their figures, enhanced the description of their methodology, and satisfactorily clarified several methodological ambiguities. Therefore, I consider that, in its current form, the manuscript is suitable for publication.